# VIA-SD: Verification via Intra-Model Routing for Speculative Decoding

Yuchen Xian [1 2]   Yang He [3 4]   Yunqiu Xu [5]   Yi Yang [1 2]

## Abstract

Speculative decoding (SD) addresses the high inference costs of LLMs by having lightweight drafters generate candidates for large verifiers to validate in parallel. Existing draft-verify methods use binary decisions: accept or fully recompute. Yet we find that many rejected tokens can be verified correctly by a slim submodel derived from the full verifier via intra-model routing, instead of the full verifier. This motivates our slim-verifier to handle tokens requiring moderate verification resources, reducing expensive large-model calls. We propose **V**erification via **I**ntr**a**-Model Routing for **S**peculative **D**ecoding (VIA-SD), a multi-tier framework using a routed slim-verifier. Draft tokens are processed hierarchically: direct acceptance for high-confidence cases, slim-verifier regeneration for medium-confidence cases, and full-model verification for uncertain cases. Across four representative tasks and multiple model families, VIA-SD reduces rejection rates by 0.10–0.22 and delivers 10–20% speedups over strong SD baselines, while achieving 2.5–3$\times$ acceleration over non-drafting decoding. Moreover, VIA-SD is compatible with existing SD frameworks without modifying their training procedures. Our results suggest multi-tier SD as a general paradigm for scalable and efficient LLM inference. Project page: https://zju-xyc.github.io/VIA-SD-Project-Page/

## 1. Introduction

Due to the high computational cost and latency involved in deploying large models for inference (Patterson, 2004;

[1]ReLER, The State Key Lab of Brain Machine Intelligence, Zhejiang University [2]College of Artificial Intelligence, Zhejiang University [3]CFAR, Agency for Science, Technology and Research, Singapore [4]IHPC, Agency for Science, Technology and Research, Singapore [5]National University of Singapore. Correspondence to: Yi Yang <yangyics@zju.edu.cn>.

*Proceedings of the 43rd International Conference on Machine Learning*, Seoul, South Korea. PMLR 306, 2026. Copyright 2026 by the author(s).

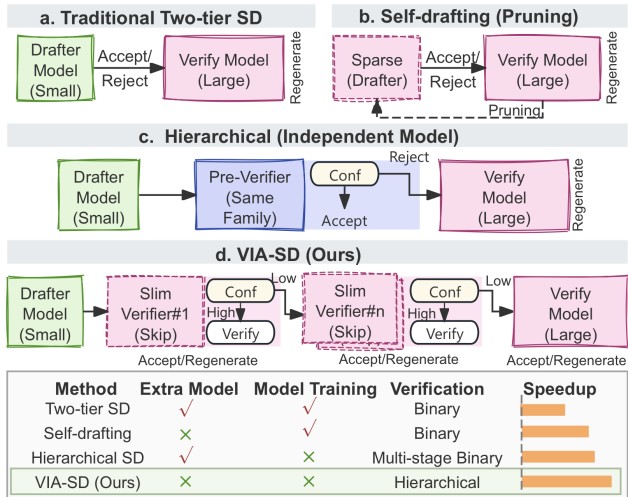

*Figure 1.* Comparison of speculative decoding (SD) paradigms. Our VIA-SD introduces a slim-verifier routed from the large model to enable hierarchical verification with low overhead.

Hennessy & Patterson, 2011; Shazeer, 2019), researchers have explored methods to improve inference efficiency at the data (Wang et al., 2018; Mirzasoleiman et al., 2020; He et al., 2024; Zhao et al., 2025), model (Shazeer et al., 2017; Han et al., 2016; Hinton et al., 2015; Zhou et al., 2025), and system levels (Xia et al., 2023; Leviathan et al., 2023; Narayanan et al., 2021; Zhou et al., 2026). At the system level, speculative decoding (SD) (Burton, 2012; Xia et al., 2023; Leviathan et al., 2023; Zhang et al., 2024) typically reduces inference cost by amortizing large-model forward passes through a two-tier *draft–verify* paradigm (see Figures 1a and 1b): a lightweight model drafts tokens, while a large model performs parallel verification and falls back to full decoding only at rejected positions.

Within the two-tier paradigm, most efforts to improve speculative decoding have focused on strengthening the drafter or accelerating the verifier (Kim et al., 2023; Zhou et al., 2024; Liu et al., 2024; Miao et al., 2024; Cai et al., 2024; Li et al., 2024a). Yet these advances largely remain within the same binary allocation rule: each token is either accepted from the small model or rejected and re-generated by the large model. Even for concurrent hierarchical SD (Syu & Lee, 2025), it relies on intermediate models to perform binary decision-making (see Figure 1c). In practice, however, we observe

that many tokens fall into a *middle zone* where the drafter is close but not fully reliable, and thus do not require the full computational capacity of the largest model to *generate*. This observation motivates a different form of **Hierarchical Verification**, in which lightweight intermediate verifiers not only assess confidence but also actively participate in token generation for middle-zone tokens, while the full model is reserved for only the hardest cases.

Building on this perspective, we reformulate speculative decoding as a staged multi-tier verification process that progressively allocates generation responsibility to increasingly stronger verifiers, rather than enforcing a rigid binary accept–reject rule. *To ground this reformulation in theory*, we analyze speculative decoding from an information-theoretic perspective and observe that its acceleration potential depends on the alignment between the drafter and verifier distributions (Narasimhan et al., 2025). While existing two-tier approaches implicitly assume a direct alignment and improve efficiency by shortening the divergence, the KL divergence (Kullback & Leibler, 1951) admits a projection geometry that noffers a useful design lens for the introduction of intermediate distributions, decomposing verification into a hierarchical sequence of stages. *For implementation*, we propose **V**erification via **Intr**a-Model Routing for **S**peculative **D**ecoding (VIA-SD) as a multi-tier verification paradigm, consisting of a drafter, slim-verifiers, and a full verifier. The slim-verifier is constructed as a routed intra-model derived from the large model, serving as an approximation to the intermediate distribution implied by the KL-based formulation. In VIA-SD, tokens drafted by the small model are first verified in parallel by the slim-verifier; upon rejection, the slim-verifier either regenerates the token itself or defers it to the full verifier, thereby substantially reducing the invocation frequency of the large model. In summary, the main contributions of this paper are as follows:

(i) We use a KL-based information-theoretic view as a design principle for introducing intermediate verification stages into the draft–verify mechanism, motivating a practical move beyond the conventional two-tier paradigm (§3.1-3.2).

(ii) We design an intermediate slim-verifier as a dynamically routed intra-model that approximates the large model, efficiently handling medium-confidence tokens and reducing redundant full-model calls (§3.3).

(iii) We propose VIA-SD as a practical multi-tier framework, and empirically identify a three-tier instantiation that balances efficiency and accuracy, alleviating the binary limitation of traditional draft–verify methods (§3.5).

(iv) Through extensive experiments on summarization,

translation, reasoning, QA and coding tasks, our proposed VIA-SD achieves consistently lower rejection rates (0.1–0.22) and delivers $10$–$20\%$ speedup over state-of-the-art speculative decoding baselines (§4).

## 2. Related Work

**Drafting Strategies.** Drafting methods broadly fall into independent and adaptive drafters. Independent drafters rely on small or non-autoregressive models (Xia et al., 2023; Leviathan et al., 2023), represented by designs such as SpecInfer and Sequoia (Miao et al., 2024; Chen et al., 2024a).Adaptive drafters reuse the target model's structure, either via auxiliary heads for parallel generation (Stern et al., 2018; Cai et al., 2024) or through layer skipping and early exiting to form lightweight submodels (Zhang et al., 2024; Yang et al., 2024).

**Verification Strategies.** Early speculative decoding adopts strict, lossless verification (Stern et al., 2018; Xia et al., 2023), guaranteeing equivalence to the target model but incurring rollback overhead. Later work introduces lossy or cascading verification (Leviathan et al., 2023; Zhou et al., 2024; Chen et al., 2024b; Narasimhan et al., 2025), relaxing acceptance or embedding deferral policies to improve efficiency. Token-tree-based verification further parallelizes validation across candidate paths (Miao et al., 2024; Cai et al., 2024; Li et al., 2024a). Overall, many recent verification-side methods rely on learned components or additional training to stabilize cross-tier decisions. Representative examples include the EAGLE series (Li et al., 2024a;b; 2025), which strengthens drafting–verification interactions, and PEARL (Liu et al., 2025), which formulates speculative decoding as a learned verification policy. While effective, these approaches trade increased training cost and tighter drafter–verifier coupling for higher runtime efficiency.

**Our Perspective.** In contrast, our work targets verification-side acceleration while adopt a distributional geometry perspective and introduce intermediate, intra-model routed verifiers that enable hierarchical verification. This design reduces rejection while preserving modularity between the drafter and verifier models.

## 3. Methodology

### 3.1. Preliminaries: Framing the Draft–Verify Paradigm

**Draft–Verify and Lossy Acceptance.** Speculative decoding follows a draft–verify paradigm, where a lightweight drafter proposes candidate tokens that are verified by a larger target model (Stern et al., 2018; Leviathan et al., 2023; Xia et al., 2023). At step $t$, the drafter samples a block of $\gamma$ tokens from $p_t(\cdot \mid x_{<t})$, which are sequentially checked under $q_t(\cdot \mid x_{<t})$. Decoding stops at the first rejection and resumes from the updated prefix; if all are accepted,

the entire block is emitted.

In probabilistic (lossy) speculative decoding (Tran-Thien, 2023), each drafted token is accepted according to relative likelihoods under $p_t$ and $q_t$, controlled by a tolerance parameter $\alpha \in [0, 1)$. Rejected tokens are replaced by samples from a residual distribution aligned with $q_t$ (details in Appendix §A). The per-step rejection rate is given by

$$\rho_t(\alpha) = \sum_{v \in \mathcal{V}} \max\{0,\, q_t(v) - (1 - \alpha)p_t(v)\}. \quad (1)$$

**Rejection Rate as a Distributional Measure.** In the lossless setting ($\alpha = 0$), the rejected probability mass $\rho_t$ corresponds to the non-overlapping support between $p_t$ and $q_t$, and can be characterized by the total variation distance:

$$\rho_t = D_{\mathrm{TV}}(p_t, q_t). \quad (2)$$

This shows that SD efficiency is fundamentally governed by the distributional gap between the drafter and the verifier: the closer $p_t$ and $q_t$ are, the lower the rejection rate.

**Limitation of TV Distance Measurement.** The total variation (TV) distance provides an exact characterization of the rejection rate in lossless SD (Villani et al., 2008):

$$D_{\mathrm{TV}}(p, q) = \tfrac{1}{2} \sum_{v \in \mathcal{V}} |p(v) - q(v)|. \quad (3)$$

In this case ($\alpha = 0$), the single-step rejection rate satisfies $\rho_t = D_{\mathrm{TV}}(p_t, q_t)$, establishing a strict equivalence that implies any reduction in rejection can only be achieved by modifying the drafter or verifier distributions themselves.

However, this perspective overlooks a critical condition: *many drafted tokens lie in an intermediate regime between $p_t$ and $q_t$.* Such *middle-zone* tokens do not require the full computational capacity of the largest verifier, yet a TV-based accept–reject decision leaves the full verifier as the only mechanism to handle them.

> **Research Question ❶:** *Can we handle middle-zone tokens by introducing a sequence of intermediate verification stages, without increasing the distributional gap?*

## 3.2. Introducing Intermediate Verifiers via Intra-Model Routing

**KL Geometry Enables Multi-Step Verification Paths.** Given the limitations of TV distance, we adopt a more general information–geometric perspective based on KL divergence (Kullback & Leibler, 1951):

$$D_{\mathrm{KL}}(p\|q) = \sum_{v \in \mathcal{V}} p(v) \log \frac{p(v)}{q(v)}. \quad (4)$$

Intuitively, $D_{\mathrm{KL}}(p\|q)$ measures the expected extra log-loss incurred when samples drawn from $p$ are encoded using $q$ rather than $p$ itself. Unlike TV distance, which provides a symmetric, global discrepancy, KL divergence quantifies a directional and additive mismatch, making it amenable to staged decomposition. A detailed comparison with TV, JS, and Wasserstein distances is provided in Appendix §C.

Let $S$ be a non-empty closed convex set on the probability simplex, and consider a sequence of intermediate distributions $\{u_i\}_{i=1}^n$, where each $u_i$ is defined via an information projection from its predecessor: $u_i = \arg\min_{u \in S} D_{\mathrm{KL}}(u_{i-1}\|u)$, with $u_0 = p$. According to the generalized Pythagorean theorem (Banerjee et al., 2005) in information geometry, the overall KL divergence between $p$ and $q$ can be decomposed as

$$D_{\mathrm{KL}}(p \,\|\, q) \geq \sum_{i=0}^{n} D_{\mathrm{KL}}(u_i \,\|\, u_{i+1}), \quad (5)$$
$$\text{with } u_0 = p,\ u_{n+1} = q,\ q \in S.$$

This inequality suggest that, within the KL framework, a multi-step path $p \to u_1 \to u_2 \to \cdots \to u_n \to q$ can yield a lower cumulative divergence than the direct mapping $p \to q$. From this perspective, intermediate distributions are not auxiliary heuristics, but principled anchors for constructing additional verification stages. Appendix §D further formalizes the family of beneficial intermediate distributions and the corresponding piecewise KL path, showing that the intermediate stage is not tied to a unique distribution $u$. The answer to RQ ❶ lies in this KL-based decomposition, which shows that middle-zone tokens need not be handled through a single binary verification step, but can instead be allocated across intermediate verification stages.

**Modeling Intermediate Distributions via Π-Space.** In practical speculative decoding, the continuous projections $u$ are not explicitly available or computationally tractable. To operationalize this idea, we reformulate the lossy mechanism through a global *target distribution* that governs the interaction between the models. Instead of comparing $p_t$ and $q_t$ directly at each step, we define a mixture target distribution:

$$\pi_t(v) := (1 - \delta)\, p_t(v) + \delta\, q_t(v), \quad (6)$$

where $\delta \in [0, 1]$ controls the weighting between the drafter and verifier. Here, $\Pi = \{\pi_t\}_{t=1}^L$ denotes the collection of target distributions across all decoding steps, and $\pi_t$ is the local realization at each step. This lossy construction defines a global $\Pi$-space, within which the conditional distributions $\{p_t, u_t, q_t\}$ remain consistent under the decoding process and provide the distributional basis for routing tokens across progressively stronger verification stages.

**When an Intermediate Verifier Is Beneficial.** On this basis, we can formalize when introducing an intermediate verifier $u$ is preferred over the direct two-tier path. For

the direct path $p \to q$, the lossy cost is $C_{\alpha,\beta}^{\mathrm{KL}}(q\|p\,|\,\pi)$. For the routed path $p \to u \to q$, the total cost decomposes into two segments, $C_{\alpha,\beta}^{\mathrm{KL}}(u\|p\,|\,\pi) + C_{\alpha,\beta}^{\mathrm{KL}}(q\|u\,|\,\pi)$, and their difference gives

$$\Delta_{\alpha,\beta}^{\mathrm{KL}}(u\,|\,\pi) = C_{\alpha,\beta}^{\mathrm{KL}}(q\|p\,|\,\pi) - C_{\alpha,\beta}^{\mathrm{KL}}(u\|p\,|\,\pi) - C_{\alpha,\beta}^{\mathrm{KL}}(q\|u\,|\,\pi). \tag{7}$$

The quantity $\Delta_{\alpha,\beta}^{\mathrm{KL}}(u\,|\,\pi)$ is the key criterion: if it is positive, the routed path is strictly better under the lossy rule in the $\pi$-space, meaning that inserting $u$ reduces the overall log-margin violation; if it is zero, the two paths are equivalent; and if it is negative, $u$ should be discarded. Empirically, we also find that the benefit is not monotonic in the number of tiers: as shown in Table 1, a three-tier setup consistently yields the best trade-off. Therefore, we instantiate our framework primarily with one intermediate verifier, which is later implemented as a slim verifier obtained through intra-model routing.

Formally, the induced target distribution in the presence of $u$ is $\pi_t^{(u)}(v) := (1 - \delta_2)\big((1 - \delta_1)\,p_t(v) + \delta_1\,u_t(v)\big) + \delta_2\,q_t(v)$, with $\delta_1, \delta_2 \geq 0$, where the coefficients are induced by the lossy thresholds $(\alpha, \beta)$ and reflect the token proportions routed through each stage.

> **Research Question ❷:** *How can we design a practical intermediate verifier via intra-model routing that effectively reduces the KL-based verification cost compared to the direct $p \to q$ path?*

### 3.3. Constructing Slim-Verifier

**Acceptance Rule.** Specifically, when the large model $q$ verifies tokens drafted by the small model $p$, the acceptance condition in speculative decoding can be expressed as

$$q(x) \geq (1 - \alpha)p(x) \iff \log \frac{q(x)}{p(x)} \geq \log(1 - \alpha) \tag{8a}$$

$$p(x) \leq \tfrac{1}{\beta}q(x) \iff \log \frac{q(x)}{p(x)} \geq \log \beta. \tag{8b}$$

These two thresholds can be unified as inequalities of log-likelihood ratios, and their violations define the *margin violation*. To quantify this deviation, we introduce a convex positive-part function $\phi : \mathbb{R} \to \mathbb{R}_{\geq 0}$—using $\phi(z) = \max\{0, z\}$ for a hard margin or $\phi(z) = \log(1 + e^z)$ for a smooth differentiable surrogate. Please refer to Appendix §F for more detailed discussions.

**Computable KL-Style Cost for Verification.** Based on this formulation, we define the KL-style single-step cost. For the direct case where $p$ drafts and $q$ verifies, the cost is

$$R_{\alpha,\beta}^{\mathrm{KL}}(q\|p) = \mathbb{E}_p\left[\phi\left(\log \frac{(1 - \alpha)p}{q}\right)\right] + \mathbb{E}_q\left[\phi\left(\log \frac{\beta p}{q}\right)\right]. \tag{9}$$

*Table 1.* Performance comparison on WebQuestions, NaturalQA, and TriviaQA with multi-layer speculative decoding. Two-layer is the speculative baseline.

| Model | # Layer | WebQuestions | | NaturalQA | | TriviaQA | |
|---|---|---|---|---|---|---|---|
| | | r | $\tau$ | r | $\tau$ | r | $\tau$ |
| Gemma2 2B→9B | 2 | 0.17 | 1.50× | 0.27 | 1.32× | 0.21 | 1.55× |
| | 3 | **0.10** | **1.82×** | **0.17** | **2.10×** | **0.13** | **2.33×** |
| | 4 | 0.13 | 1.72× | 0.21 | 1.84× | 0.16 | 1.93× |
| | 5 | 0.29 | 0.89× | 0.37 | 0.98× | 0.31 | 0.94× |
| Gemma2 2B→27B | 2 | 0.27 | 1.55× | 0.45 | 1.54× | 0.24 | 1.65× |
| | 3 | **0.14** | **2.32×** | **0.30** | **2.61×** | **0.15** | **2.50×** |
| | 4 | 0.19 | 2.05× | 0.36 | 2.18× | 0.20 | 2.10× |
| | 5 | 0.32 | 1.01× | 0.51 | 0.99× | 0.29 | 1.05× |

The first term penalizes cases where $q$'s support for $p$ falls short of the $(1 - \alpha)$ threshold, corresponding exactly to the *log-margin violation of the acceptance threshold* in the original decomposition. The second term accounts for the log-margin cost of residual replacement, *i.e.*, the *log-margin contribution of residual replacement* that arises when $q$ must "borrow" probability mass to replace $p$'s draft. In an extended multi-tier structure, if $u$ first verifies $p$, we similarly obtain $R_{\alpha,\beta}^{\mathrm{KL}}(u\|p)$; and if $q$ subsequently verifies $u$, we obtain $R_{\alpha,\beta}^{\mathrm{KL}}(q\|u)$. Thus, each segment of the multi-tier pipeline can be formalized through the same log-margin perspective.

To align with sequence generation, this single-step cost must be accumulated across the decoding time steps $\mathcal{T}$. In the $\Pi$-space induced by the lossy mechanism in Eq. (6), the block-level cost is

$$C_{\alpha,\beta}^{\mathrm{KL}}(q\|p\,|\,\pi) = \sum_{t \in \mathcal{T}} R_{\alpha,\beta}^{\mathrm{KL}}\big(q_t\|p_t\big) \tag{10}$$
$$\text{with } q_t, p_t \in \Delta_V \text{ induced by } \pi.$$

Here, $\pi$ ensures consistency among the conditional distributions $\{p_t, q_t, u_t\}$ across different stages. However, in the definition of $C_{\alpha,\beta}^{\mathrm{KL}}(q\|p\,|\,\pi)$, the expectations involved cannot be computed analytically during inference, and thus must be transformed into a computable form. For example, let $\ell^p = \log p_t(\cdot)$ and $\ell^q = \log q_t(\cdot)$, and define $z_1(v) = \log(1 - \alpha) + \ell^p(v) - \ell^q(v)$, $z_2(v) = \log \beta + \ell^p(v) - \ell^q(v)$. These quantities convert the two acceptance conditions into token-wise log-margin violations, making the original distributional cost evaluable from model logits. For a simple, inference-time computable surrogate aligned with the accept/reject thresholds, we choose the convex positive-part $\phi(z) = \max\{0, z\}$ (ReLU), which penalizes margin violations and ignores satisfied constraints; a smooth alternative (*e.g.*, $\log(1 + e^z)$) is discussed in Appendix §F.

When choosing $\phi(z) = \max\{0, z\}$ (*i.e.*, ReLU), the KL-

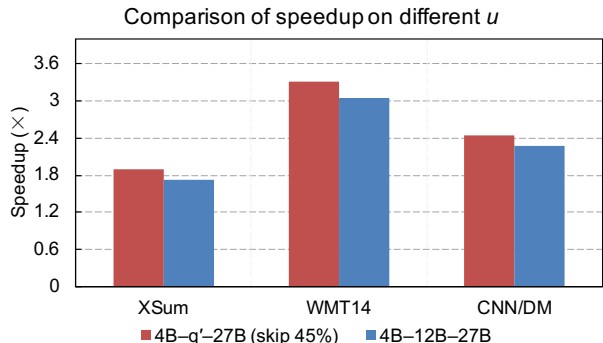

*Figure 2.* Skip-layer intermediate consistently outperforms the independent 12B model on Gemma3 pilot study.

style lossy cost for a decoding step can be written as

$$
\begin{aligned}
R^{\mathrm{KL}}_{\alpha,\beta}(q\|p)\big|_t = & \\
\underbrace{\sum_v p_t(v)\,\mathrm{ReLU}(z_1(v))}_{\text{acceptance threshold term}} & + \underbrace{\sum_v q_t(v)\,\mathrm{ReLU}(z_2(v))}_{\text{residual replacement term}}.
\end{aligned}
\tag{11}
$$

This expression converts the original log-margin violation and residual replacement requirements into a token-wise computable sum. By further summing over all decoding steps $t \in \mathcal{T}$, the block-level cost is obtained as $C^{\mathrm{KL}}_{\alpha,\beta}(q\|p \mid \pi) = \sum_{t\in\mathcal{T}} R^{\mathrm{KL}}_{\alpha,\beta}(q\|p)\big|_t$, thereby making Eq. (10) computable in practice. The detailed derivation is provided in Appendix §G.

**Identifying the Slim-Verifier.** To operationalize the intermediate verifier $u$, we consider three broad candidates: (i) a scaled-up variant $p'$ of the drafter $p$, (ii) an independent smaller model from the same family as the verifier $q$, and (iii) a routed submodel $q'$ derived from $q$. We adopt the routed intra-model $q'$.

The key reason is distributional consistency. Although $q'$ has a parameter scale comparable to that of an independent intermediate model, it shares embeddings and the output head with the full verifier $q$. As a result, when substituting $u = q'$ into Eq. (11), the block-level cost $C^{\mathrm{KL}}_{\alpha,\beta}(q\|q' \mid \pi)$ is consistently smaller than that of an independent model of equal size. Consequently, the routed path $p \rightarrow q' \rightarrow q$ is more likely to satisfy $\Delta^{\mathrm{KL}}_{\alpha,\beta}(u \mid \pi) > 0$, leading to lower rejection rates. By adjusting the intra-model routing ratio, this design further enables a controllable trade-off between decoding acceleration and accuracy (see Figure 2 and Appendix §E). We therefore adopt $q'$ as the intermediate verifier in our framework. This answers ⟨RQ ❷⟩ by instantiating the intermediate stage as a routed intra-model derived from $q$, which remains efficient while preserving distributional consistency with the full verifier.

**Algorithm 1** Hierarchical Verification with DIMR (simplified; the full version is in Algorithm 2 in Appendix §B)

**Require:** Drafter $p$, full verifier $q$, skip ratio $r$, block length $\gamma$, thresholds $(\alpha_1, \alpha_2)$
**Ensure:** Final decoded sequence $x$
1: Select routing mask $z^*$ by offline DIMR under ratio $r$
2: Construct fixed slim-verifier $q' \leftarrow \Pi_{z^*}(q)$
3: **while** not end-of-sequence **do**
4:   Draft a block $\tilde{x}_{t:t+\gamma-1} \sim p$
5:   Verify the drafted block with the staged gates ($p \rightarrow q'$) and ($q' \rightarrow q$)
6:   Accept the longest valid prefix $\tilde{x}_{t:t+\kappa-1}$
7:   **if** $\kappa < \gamma$ **then**
8:     Regenerate token $x_{t+\kappa}$ using $q'$ or fallback to $q$
9:     Discard the remaining suffix $\tilde{x}_{t+\kappa+1:t+\gamma-1}$
10:     Append $\tilde{x}_{t:t+\kappa-1}$ and regenerated token $x_{t+\kappa}$
11:     $t \leftarrow t + \kappa + 1$
12:   **else**
13:     Append the full accepted block $\tilde{x}_{t:t+\gamma-1}$
14:     $t \leftarrow t + \gamma$
15:   **end if**
16: **end while**

### 3.4. Dynamic Intra-Model Routing (DIMR) for Stable Slim-Verifier Selection

**Intra-Model Routing Formulation of Slim-Verifier.** Let the full verifier $q$ consist of $L$ Transformer layers. We define a routing mask as $z \in \{0,1\}^L$, where $z_\ell = 1$ indicates that the $\ell$-th layer is retained, and $z_\ell = 0$ indicates that it is bypassed under intra-model routing. Based on $z$, the slim-verifier can be formally defined as $q'_z(x) = \Pi_z(q)(x)$, where $\Pi_z(\cdot)$ denotes the projection of $q$ onto the routed subspace specified by $z$, resulting in a compact routed submodel $q'_z$.

For each candidate routing mask $z$ that defines a slim-verifier $q'_z$, we score $z$ by accumulating a KL-style per-step cost, see Eq. (11), over a context window of length $\tau$:

$$
\mathcal{C}(z) = \sum_{t=1}^{\tau} R^{\mathrm{KL}}_{\alpha,\beta}(q \| q'_z)\big|_t, \quad z^* = \arg\min_z \mathcal{C}(z). \tag{12}
$$

**Dynamic Routing Optimization.** To balance efficiency and accuracy, DIMR combines *random search* with *periodic Bayesian optimization*:

$$
z = \begin{cases} \mathrm{BayesOpt}(l), & \text{if } o \bmod \theta = 0, \\ \mathrm{RandomSearch}(l), & \text{otherwise,} \end{cases} \tag{13}
$$

where $o$ is the current optimization step, $\theta$ is the triggering period for Bayesian optimization, and $l = \binom{L}{rL}$ represents the search space of candidate routing masks (with $r$ denoting

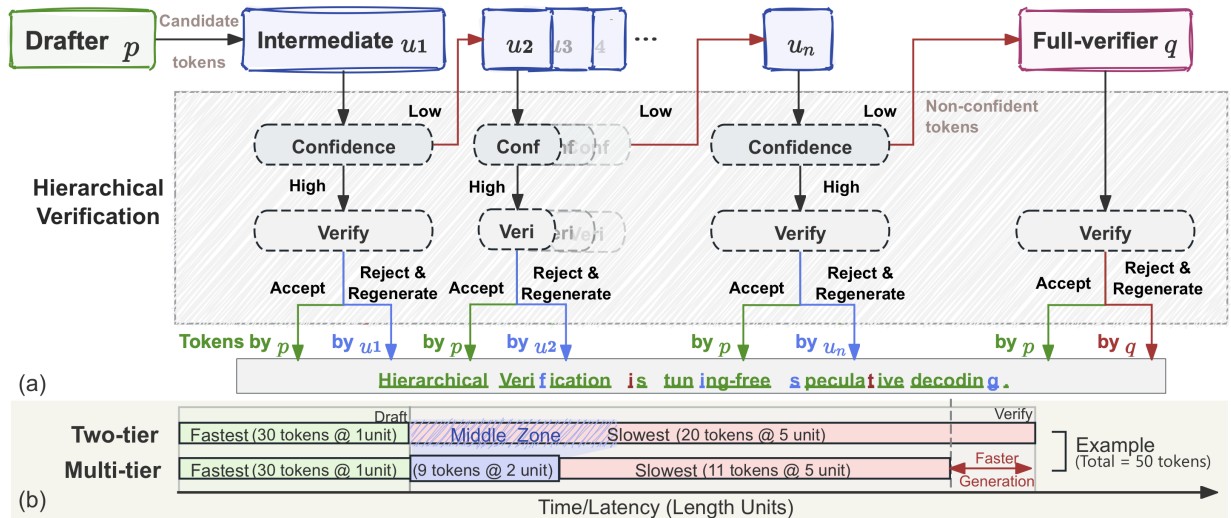

*Figure 3.* Illustration of our proposed VIA-SD pipeline. (a) Candidate tokens drafted by $p$ are progressively verified by a hierarchy of intermediate verifiers $\{u_1, \ldots, u_n\}$, where high-confidence tokens are accepted early and only the remaining uncertain tokens are deferred to the full verifier $q$. (b) Latency comparison between two-tier and hierarchical multi-tier verification, showing reduced reliance on full-model decoding.

the intra-model routing ratio). In practice, Bayesian optimization treats the evaluated mask–cost pairs $\{(z, \mathcal{C}(z))\}$ as observations and proposes promising discrete routing masks for subsequent evaluation. The intervening random-search steps maintain exploration over the combinatorial mask space, preventing the search from prematurely concentrating on locally favorable layer patterns. This strategy enables efficient exploration while progressively approaching the optimal routing configuration. The structure of $q'$ is fixed, once the maximum optimization step $S$ is reached, or the best candidate remains unchanged across multiple iterations. Then $q'$ serves as a stable approximation of $q$.

### 3.5. VIA-SD Pipeline

**VIA-SD Inference.** Figure 3 depicts the overall pipeline of our proposed VIA-SD. As illustrated in Algorithm 1, given a decoding prefix $x_{<t}$, the inference procedure proceeds as follows at each decoding cycle:

*Step 1: Block drafting.* At step $t$, the drafter $p$ proposes a block of up to $\gamma$ tokens $\tilde{x}_{t:t+\gamma-1} \sim p$, enabling amortized verification over a longer prefix.

*Step 2: Slim-verifier construction.* Under a fixed intra-model routing ratio $r$, a slim-verifier $q'$ is derived from the full verifier $q$ via Dynamic Intra-Model Routing (DIMR), yielding the approximate distribution $q'_t(v)$.

*Step 3: Hierarchical verification.* The drafted block is verified using two confidence thresholds $(\delta_1, \delta_2)$. The early gate $p \to q'$ is stricter, while the later gate $q' \to q$ is more permissive, with $\delta_1 \gg \delta_2$ in practice.

*Step 4: Prefix acceptance and fallback.* The longest ac-

cepted prefix length is $\kappa = \max_{0 \le k \le \gamma} \Big\{ k \mid \tilde{x}_{t:t+k-1} \in \mathcal{A}(p \to q', \, q' \to q) \Big\}$. If $\kappa < \gamma$, decoding falls back to the full verifier $q$ at step $t + \kappa$. The rewritten token updates the prefix, and the remaining draft suffix is discarded before the next cycle, avoiding verification under an outdated conditional generation context.

*Step 5: Mixture target distribution.* The effective token distribution is $\pi_t^{(q')}(v) = (1-\delta_2)\Big((1-\delta_1)\, p_t(v) + \delta_1\, q'_t(v)\Big) + \delta_2\, q_t(v)$, where $\delta_1$ and $\delta_2$ are induced by the confidence thresholds $(\alpha_1, \alpha_2)$.

## 4. Experiments

### 4.1. Experimental Setting

**Setups and Metrics.** We evaluate our method on both encoder–decoder and decoder-only model families, including T5 (Raffel et al., 2020), Gemma2 (Team et al., 2024), LLaMA2 (Touvron et al., 2023), and Qwen (Bai et al., 2023), pairing small-to-large configurations, covering T5-S→L, T5-L→XL, Gemma2-2B→9B/27B, LLaMA2-7B→13B/70B, and Qwen-7B→14B/72B.

Encoder–decoder models are evaluated on XSum (Narayan et al., 2018), CNN/DailyMail (See et al., 2017), and WMT14 En–De (Bojar et al., 2014), while decoder-only models are benchmarked on GSM8K (8-shot) (Cobbe et al., 2021), MBPP (Austin et al., 2021), SQuAD 2.0 (Rajpurkar et al., 2018), WebQuestions (Berant et al., 2013), Natu-ralQA (Kwiatkowski et al., 2019), and TriviaQA (Joshi et al., 2017). We report task-specific quality metrics and

*Table 2.* Performance comparison of results on WebQuestions, NaturalQA, and TriviaQA benchmarks, where *Acc.*, *Rej.* and *Speed* denote Accuracy, Rejection Rate and Speedup metrics, respectively.

| Model | Method | WebQuestions | | | NaturalQA | | | TriviaQA | | |
|---|---|---|---|---|---|---|---|---|---|---|
| | | Acc. | Rej. | Speed | Acc. | Rej. | Speed | Acc. | Rej. | Speed |
| Gemma2 2B→9B | Speculative Decoding | 0.27 | 0.17 | 1.50× | 0.26 | 0.27 | 1.32× | 0.50 | 0.21 | 1.55× |
| | Cascade SD | 0.27 | 0.16 | 1.53× | 0.26 | 0.26 | 1.43× | 0.50 | 0.20 | 1.62× |
| | Faster Cascades | 0.28 | 0.13 | 1.65× | 0.27 | 0.22 | 1.75× | 0.52 | 0.17 | 1.90× |
| | VIA-SD (ours) | **0.28** | **0.10** | **1.82×** | **0.28** | **0.17** | **2.10×** | **0.53** | **0.13** | **2.33×** |
| Gemma2 2B→27B | Speculative Decoding | 0.32 | 0.27 | 1.55× | 0.32 | 0.45 | 1.54× | 0.54 | 0.24 | 1.65× |
| | Cascade SD | 0.32 | 0.24 | 1.71× | 0.32 | 0.42 | 1.73× | 0.54 | 0.23 | 1.81× |
| | Faster Cascades | **0.33** | 0.22 | 1.81× | 0.33 | 0.40 | 2.10× | **0.56** | 0.20 | 2.30× |
| | VIA-SD (ours) | 0.32 | **0.14** | **2.32×** | **0.34** | **0.30** | **2.61×** | 0.55 | **0.15** | **2.50×** |
| LLaMA2 7B→13B | Speculative Decoding | 0.26 | 0.22 | 1.42× | 0.25 | 0.34 | 1.28× | 0.49 | 0.26 | 1.48× |
| | Cascade SD | 0.26 | 0.21 | 1.48× | 0.25 | 0.32 | 1.40× | 0.49 | 0.24 | 1.60× |
| | Faster Cascades | 0.27 | 0.17 | 1.62× | 0.26 | 0.28 | 1.68× | 0.51 | 0.20 | 1.82× |
| | VIA-SD (ours) | **0.27** | **0.13** | **1.88×** | **0.27** | **0.22** | **2.05×** | **0.52** | **0.16** | **2.20×** |
| LLaMA2 7B→70B | Speculative Decoding | 0.31 | 0.30 | 1.50× | 0.31 | 0.48 | 1.45× | 0.54 | 0.27 | 1.60× |
| | Cascade SD | 0.31 | 0.27 | 1.72× | 0.31 | 0.46 | 1.65× | 0.54 | 0.26 | 1.80× |
| | Faster Cascades | **0.32** | 0.24 | 1.85× | 0.32 | 0.42 | 2.00× | **0.56** | 0.22 | 2.15× |
| | VIA-SD (ours) | 0.31 | **0.16** | **2.30×** | **0.33** | **0.33** | **2.55×** | 0.55 | **0.18** | **2.45×** |
| Qwen 7B→14B | Speculative Decoding | 0.31 | 0.20 | 1.50× | 0.31 | 0.32 | 1.35× | 0.54 | 0.24 | 1.58× |
| | Cascade SD | 0.31 | 0.19 | 1.58× | 0.31 | 0.30 | 1.50× | 0.54 | 0.23 | 1.70× |
| | Faster Cascades | 0.32 | 0.15 | 1.72× | 0.32 | 0.26 | 1.80× | 0.56 | 0.19 | 1.95× |
| | VIA-SD (ours) | **0.33** | **0.11** | **2.05×** | **0.33** | **0.20** | **2.25×** | **0.57** | **0.15** | **2.40×** |
| Qwen 7B→72B | Speculative Decoding | 0.36 | 0.28 | 1.60× | 0.36 | 0.47 | 1.55× | 0.60 | 0.26 | 1.70× |
| | Cascade SD | 0.36 | 0.25 | 1.85× | 0.36 | 0.45 | 1.70× | 0.60 | 0.25 | 1.90× |
| | Faster Cascades | **0.37** | 0.22 | 2.00× | 0.37 | 0.41 | 2.15× | **0.62** | 0.21 | 2.25× |
| | VIA-SD (ours) | 0.37 | **0.15** | **2.45×** | **0.38** | **0.32** | **2.70×** | 0.61 | **0.17** | **2.55×** |

*Table 3.* Performance comparison (Xsum, CNNDM, and WMT14) under different decoding settings.

| Model | Method | XSum (T=1) | | | CNNDM (T=1) | | | WMT14 (T=0) | | |
|---|---|---|---|---|---|---|---|---|---|---|
| | | ROUGE-2 | Rej. | Speed | ROUGE-2 | Rej. | Speed | BLEU | Rej. | Speed |
| T5 S→L | Speculative Decoding | 14.80 | 0.36 | 1.02× | 11.20 | 0.36 | 1.79× | 18.00 | 0.30 | 1.40× |
| | Cascade SD | 14.80 | 0.33 | 1.15× | 11.20 | 0.33 | 1.90× | 18.00 | 0.28 | 1.60× |
| | Faster Cascades | 15.05 | 0.30 | 1.30× | 12.63 | 0.34 | 1.88× | **22.65** | 0.25 | 1.85× |
| | VIA-SD (ours) | **15.27** | **0.24** | **1.90×** | **12.81** | **0.26** | **2.10×** | 21.92 | **0.22** | **2.50×** |
| T5 S→XL | Speculative Decoding | 18.90 | 0.38 | 1.12× | 12.90 | 0.38 | 1.70× | 22.95 | 0.34 | 1.80× |
| | Cascade SD | 18.90 | 0.35 | 1.25× | 12.90 | 0.35 | 1.85× | 22.95 | 0.32 | 2.00× |
| | Faster Cascades | **18.99** | 0.30 | 1.45× | 12.95 | 0.34 | 1.95× | 23.05 | 0.25 | 2.70× |
| | VIA-SD (ours) | 18.95 | **0.28** | **1.65×** | **13.05** | **0.24** | **2.40×** | **23.10** | **0.21** | **3.35×** |

measure efficiency by rejection rate and speedup over greedy decoding. All experiments are conducted on A100 GPUs with consistent decoding settings.

**Baselines.** We compare against representative (1) *independent drafting methods* (Speculative Decoding (Leviathan et al., 2023) and BiLD (Kim et al., 2023)), (2) *lossy cascade-based approaches* (Cascade Speculative Drafting (Chen et al., 2024b) and Faster Cascades (Narasimhan et al., 2025)), and (3) *self-drafting methods* (Swift (Xia et al., 2025) and CLaSP (Chen et al., 2025)). Due to space constraints, we report results for SD, Cascade SD, and Faster Cascades in the main text, while the other baselines are deferred to Appendix §H.1.

**Hyperparameters.** We set $\alpha_1 = 0.5$, $\alpha_2 = 0.3$, $\gamma = 5$,

and use a 45% intra-model routing ratio to construct the slim-verifier. These values reflect a conservative separation between intermediate and full verification, balancing early handling of middle-zone tokens against controlled escalation to the full verifier.

### 4.2. Main Results

**Decoder-Only Models.** Across QA and reasoning benchmarks (Table 2), our intra-model routing-based hierarchical verification reduces rejection rates by roughly 30–45% relative to speculative decoding, leading to consistent speedup improvements of 0.3×–0.8× over the strongest cascade baselines, while maintaining or slightly improving accuracy. The effect scales with the capacity gap between the

*Table 4.* One-time offline DIMR search cost for each model pair.

| Model | Wall-clock Time | GPU-hours |
|---|---|---|
| Gemma2 2B→9B | 18 min | 0.30 h |
| Gemma2 2B→27B | 34 min | 0.57 h |
| LLaMA2 7B→13B | 26 min | 0.43 h |
| LLaMA2 7B→70B | 61 min | 1.02 h |
| Qwen 7B→14B | 29 min | 0.48 h |
| Qwen 7B→72B | 68 min | 1.13 h |

*Table 5.* Controlled ablation of intermediate verifier construction on WebQuestions with Gemma2 2B→27B.

| Setting | Extra model | Peak mem. | Speed | Acc. |
|---|---|---|---|---|
| Two-layer SD | ✗ | **1.00×** | 1.55× | 0.32 |
| Independent 13B | ✔ | 1.38× | 2.08× | **0.33** |
| Random skip-layer | ✗ | 1.04× | 1.62× | 0.29 |
| DIMR-routed (ours) | ✗ | 1.04× | **2.32×** | 0.32 |

*Table 6.* Ablation on intermediate construction by replacing DIMR with random skip-layer selection under the same hierarchical verification pipeline and skip ratio.

| Model | Skip Method | WebQ | | NatQA | | TriviaQA | |
|---|---|---|---|---|---|---|---|
| | | Acc. | Speed | Acc. | Speed | Acc. | Speed |
| Gemma2 2B→9B | Random | 0.26 | 1.32× | 0.25 | 1.41× | 0.50 | 1.58× |
| | Ours | **0.28** | **1.82×** | **0.28** | **2.10×** | **0.53** | **2.33×** |
| Gemma2 2B→27B | Random | 0.29 | 1.47× | 0.30 | 1.65× | 0.54 | 1.73× |
| | Ours | **0.32** | **2.32×** | **0.34** | **2.61×** | **0.55** | **2.50×** |

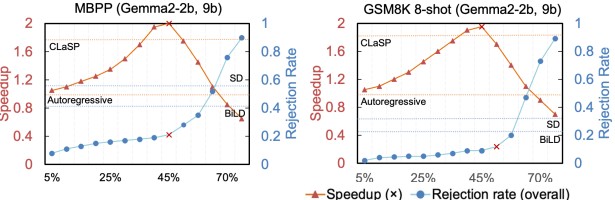

*Figure 4.* Ablation results of skip ratio on Gemma2-2B and 9B.

drafter and verifier. In large-gap settings (*e.g.*, 2B→27B or 7B→70B), rejection rates decrease from approximately 0.22–0.30 to 0.14–0.16, enabling speedups above 2.3×–2.7×. These results confirm that decoder-only generation contains a substantial "middle zone" of uncertain tokens that are poorly handled by binary accept–reject verification but can be efficiently resolved through staged verification with an intra-model routed intermediate verifier. Detailed experiments are deferred to Appendix §H.

**Encoder-Decoder Models.** Across XSum and CNN/DailyMail on encoder–decoder models (Table 3), our method reduces rejection rates by approximately 8–12 percentage points compared to speculative decoding and cascade-based baselines, resulting in an additional 1.2×–1.4× speedup without degrading ROUGE scores. This suggests that for summarization tasks, where token uncertainty is relatively concentrated, most drafted tokens can be resolved at early routed verification stages. The advantage becomes more pronounced on WMT14. Our intra-model routing-based hierarchical verification lowers rejection rates from around 0.25–0.30 to 0.21–0.22, yielding speedups up to 2.5×–3.3×, substantially higher than existing cascade methods. This indicates that the routed intermediate verifier is particularly effective at filtering low-confidence translation tokens before invoking full-model decoding. Detailed results are provided in Appendix §H.

**Search Cost.** Since VIA-SD relies on DIMR to construct the routed slim-verifier, we further separate the offline routing-mask search cost from the online decoding speedups reported above. DIMR is executed once for each model pair to obtain a fixed slim-verifier, and no routing-mask search is performed during inference. As shown in Table 4, the one-time search cost remains moderate across model families, ranging from 18 to 68 minutes, or 0.30 to 1.13 GPU-hours. Although the cost increases with the size of the full verifier, it is incurred only once and can be reused across down-

stream tasks under the same model pair. Thus, the speedups reported in Tables 2 and 3 correspond to online decoding acceleration rather than amortized end-to-end speedup including offline search.

### 4.3. Ablation Studies

**Slim-Verifier Construction.** Table 5 disentangles the source of VIA-SD's improvement. Compared with two-layer speculative decoding, introducing an independent 13B intermediate model improves speed from 1.55× to 2.08× and slightly increases accuracy from 0.32 to 0.33. However, this design requires loading an additional model and increases peak memory to 1.38×, which weakens its practicality in memory-constrained deployment. Random skip-layer routing avoids extra model loading and incurs only a small memory overhead, but it provides marginal acceleration and reduces accuracy, suggesting that merely inserting a third stage is insufficient. In contrast, DIMR-routed slim-verifier achieves the best trade-off, reaching 2.32× speedup with only 1.04× peak memory and without accuracy degradation.

**Skip Ratio.** As shown in Table 6, replacing DIMR with random routing masks consistently reduces both accuracy and speedup, indicating that the intermediate verifier must be carefully selected rather than randomly constructed. We further study the intra-model routing ratio in Figure 4. Speedup first improves as more layers are skipped, but drops when the ratio becomes too large, since an overly weak slim-verifier triggers more fallbacks. This suggests a moderate routing regime that balances efficiency and verification reliability.

**Threshold Robustness.** Table 7 studies the sensitivity of VIA-SD to threshold and skip-ratio choices. The default setting achieves the best average speedup on both Gemma2 2B→9B and 2B→27B, while both conservative and aggres-

*Table 7.* Robustness of VIA-SD under different confidence thresholds and intra-model routing ratios. Results are averaged over WebQuestions, NaturalQA, and TriviaQA.

| Model | Setting | $(\alpha_1, \alpha_2)$ | Skip Ratio | Speed |
|---|---|---|---|---|
| Gemma2 2B→9B | Conservative | (0.4, 0.2) | 35% | 1.71× |
| | Default (ours) | (0.5, 0.3) | 45% | **2.08×** |
| | Aggressive | (0.6, 0.4) | 55% | 1.93× |
| Gemma2 2B→27B | Conservative | (0.4, 0.2) | 35% | 2.18× |
| | Default (ours) | (0.5, 0.3) | 45% | **2.48×** |
| | Aggressive | (0.6, 0.4) | 55% | 2.27× |

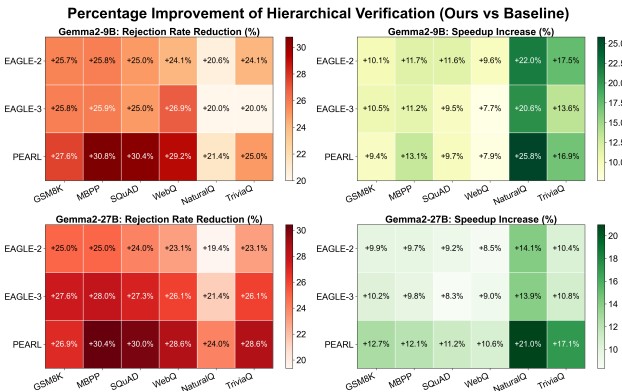

*Figure 5.* Percentage improvement of hierarchical verification over baseline speculative decoding methods. Cells report the relative gain of *Method+Ours* over the baseline in rejection and decoding speedup for Gemma2-9B and Gemma2-27B.

sive settings remain competitive. This suggests that VIA-SD degrades smoothly under moderate hyperparameter changes, but an overly conservative or aggressive routing policy can reduce the efficiency gain.

### 4.4. Additional Analysis and Discussions

**Why Does Multi-Tier Verification Improve Generation Quality?** Three-level verification can improve task-level quality and sometimes even outperform direct decoding with the target model $q$. This gain is a consequence of the lossy nature of SD: inference-time gating reshapes the effective decoding distribution, introducing a small but useful deviation that can better align with sequence-level metrics than strict token-level likelihood under $q$.

The induced per-step distribution is the gated mixture

$$\pi_t^{(q')}(v) = (1 - \delta_2)\Big((1 - \delta_1)p_t(v) + \delta_1 q'_t(v)\Big) + \delta_2 q_t(v),$$
(14)

where $\delta_1 = \delta_1(\alpha_1)$ and $\delta_2 = \delta_2(\alpha_2)$ are gating indicators induced by the confidence thresholds. Unlike lossless SD (which preserves $q$), Eq. (14) defines a *lossy re-weighting* over $\{p, q', q\}$.

Our intra-model routing-based hierarchical design makes this lossy deviation *stable*: $q'$ handles medium-confidence states with soft correction, while $q$ is reserved for low-confidence fallbacks. Equivalently, it reduces brittle two-mode behavior by allocating more probability mass to $q'$ when $p$ is unreliable but escalation to $q$ is not necessary, *i.e.*, $\mathbb{E}[\delta_2] \downarrow$ and $\mathbb{E}[\delta_1] \uparrow$ (relative to a two-tier $p \to q$ gate), thereby lowering catastrophic deviations while retaining the beneficial exploration of lossy SD, which yields consistent quality gains. See more discussions in Appendix §H.2.

**Why Is Our Method Compatible with Existing Draft–Verify Frameworks?** As shown in Figure 5, VIA-SD can be directly integrated into representative speculative decoding systems, including the EAGLE series (Li et al., 2024a;b; 2025) and PEARL (Liu et al., 2025). Across Gemma2-9B and Gemma2-27B backbones, adding our hierarchical verifier (+Ours) consistently improves these baselines, reducing rejection rates by roughly 20–31% and increasing decod-

ing speedup by about 8–26% across reasoning, code, and QA tasks. The gains are particularly clear on PEARL and QA/code-heavy benchmarks, where many tokens rejected by the original binary verifier can instead be resolved by the intermediate slim-verifier. This indicates that VIA-SD improves the verification stage itself rather than relying on task-specific or baseline-specific properties of a particular drafter.

This compatibility comes from preserving the standard draft–verify interface. Existing methods propose candidate tokens, accept the longest valid prefix, and fall back to the target model for uncertain tokens. VIA-SD leaves this process unchanged, but inserts a lightweight intra-model routed verifier between direct acceptance and full-model verification. It therefore refines the original binary accept–escalate decision into staged verification, while remaining applicable to both external-drafter and self-drafting pipelines without retraining or modifying the verifier.

## 5. Conclusion

This paper reframes speculative decoding as a multi-tier verification problem beyond the conventional binary draft–verify paradigm. By introducing hierarchical verification with an intra-model routed slim verifier, VIA-SD enables finer-grained allocation of verification effort across tokens of varying confidence, reducing unnecessary full-model invocations without sacrificing generation quality. From a theoretical perspective, the KL-based view motivates intermediate distributions and provides a practical design principle for staged verification. Guided by this insight, VIA-SD uses intra-model routing to construct lightweight intermediate verifiers that actively participate in token generation and verification. Together, our analysis and system demonstrate that multi-tier speculative decoding offers an effective paradigm for efficient LLM inference.

## Acknowledgments

This research is partially supported by the Fundamental and Interdisciplinary Disciplines Breakthrough Plan of the Ministry of Education of China. This research is partially supported by A*STAR Career Development Fund (CDF) under Grant C243512011, the National Research Foundation, Singapore under its National Large Language Models Funding Initiative (AISG Award No: AISG-NMLP-2024-003). Any opinions, findings and conclusions or recommendations expressed in this material are those of the author(s) and do not reflect the views of National Research Foundation, Singapore.

## Impact Statement

This paper advances efficient inference for large language models by introducing VIA-SD, a multi-tier speculative decoding framework that inserts an intra-model routed slim verifier between the drafter and the full verifier. By reducing unnecessary full-model invocations while maintaining (and sometimes slightly improving) task quality, VIA-SD can lower latency, compute cost, and energy consumption for LLM deployment, which may broaden access to capable models in resource-constrained settings and improve the sustainability of large-scale serving.

For deployment dependence, our speed measurements are conducted on A100 GPUs under controlled decoding settings. Absolute speedups may vary with batch size, sequence length, KV-cache implementation, kernel fusion, memory bandwidth, and hardware generation. Nevertheless, rejection-rate reduction reflects a model-level decrease in full-verifier invocations and remains a useful indicator of potential serving efficiency.

At the same time, faster and cheaper text generation can increase the throughput of both benign and harmful uses (*e.g.*, spam, misinformation, or automated social engineering). Our method does not introduce new data, does not alter model training, and does not directly change the underlying model's propensity to generate unsafe content; however, it can reduce the marginal cost of generating such content. We therefore recommend deploying VIA-SD together with existing safety mitigations (content filtering, rate limiting, monitoring, and abuse detection) and evaluating system-level changes under safety and robustness benchmarks before real-world release.

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

## A. Notation and Details of Lossy Speculative Decoding

**Notation.** Let $\mathcal{V}$ denote the vocabulary and $\Delta_{\mathcal{V}}$ the probability simplex over $\mathcal{V}$. At decoding step $t$, given a prefix $x_{<t}$, the drafter model $M_p$ and verifier model $M_q$ induce conditional next-token distributions $p_t(\cdot \mid x_{<t}), q_t(\cdot \mid x_{<t}) \in \Delta_{\mathcal{V}}$.

**Drafting Process.** Given a block size $\gamma$, the drafter sequentially samples

$$x_t, x_{t+1}, \ldots, x_{t+\gamma-1} \sim p_t(\cdot \mid x_{<t}), p_{t+1}(\cdot \mid x_{\leq t}), \ldots, p_{t+\gamma-1}(\cdot \mid x_{<t+\gamma-1}). \tag{A.1}$$

The verifier evaluates the drafted tokens in order and halts upon the first rejection.

**Acceptance Probability.** In lossy speculative decoding (Tran-Thien, 2023), a drafted token $x_{t+j}$ is accepted with probability

$$\Pr[\text{accept } x_{t+j}] = \min\left(1, \ \frac{q_{t+j}(x_{t+j})}{(1-\alpha)\,p_{t+j}(x_{t+j})}\right), \tag{A.2}$$

where $\alpha \in [0, 1)$ controls the allowed deviation between the drafter and verifier distributions.

**Residual Distribution.** If a drafted token is rejected, it is resampled from the residual distribution

$$x_{t+j} \sim \text{norm}\left(\max\left\{0, \ \frac{1}{\beta}q_{t+j}(\cdot) - p_{t+j}(\cdot)\right\}\right), \tag{A.3}$$

where $\beta \geq 1 - \alpha$ is a scaling parameter and $\text{norm}(\cdot)$ denotes normalization to a valid probability distribution. This construction ensures that replacement tokens remain aligned with the verifier distribution.

**Lossless Case.** When $\alpha = 0$ and $\beta = 1$, the procedure reduces to standard autoregressive sampling from $q$, recovering a lossless decoding process (Leviathan et al., 2023; Chen et al., 2023).

# B. Hierarchical Verification Pipeline: Derivation of $\delta_1$, $\delta_2$ and Their Relation to $\alpha_1$, $\alpha_2$

## B.1. Full Pseudocode for DIMR Pipeline

**Algorithm summary.** At each step, the drafter $p$ proposes a $\gamma$-token block. We derive a slim verifier $q'$ from the full verifier $q$ via DIMR (skip ratio $r$) and verify tokens in order using two thresholds: if $q'$ is confident, it accepts or rewrites; otherwise it escalates to $q$. The block is committed only up to the earliest rewrite position, after which decoding falls back to $q$ and continues.

---

**Algorithm 2** Hierarchical Verification with DIMR (aligned with §3.5)

---

**Require:** Drafter $p$, full verifier $q$, skip ratio $r$, block length $\gamma$, thresholds $(\alpha_1, \alpha_2)$
**Ensure:** Final decoded sequence $x$
1: Initialize running estimates for mixture weights $\delta_1, \delta_2$ (optional)
2: **while** not end-of-sequence **do**
3:      Draft a block $\tilde{x}_{t:t+\gamma-1} \sim p$
4:      Build slim-verifier $q'$ from $q$ via DIMR under ratio $r$
5:      Run $q'$ in parallel on prefixes $x_{<t}, \ldots, x_{<t+\gamma-1}$ to obtain $q'_{t:t+\gamma-1}(\cdot)$
6:      $j^* \leftarrow$ None          ▷ earliest rewrite/reject position
7:      **for** $j = 0$ **to** $\gamma - 1$ **do**
8:          $v \leftarrow \tilde{x}_{t+j}, \quad M' \leftarrow \max_u q'_{t+j}(u)$
9:          **if** $q'_{t+j}(v) \geq (1 - \alpha_2) M'$ **then**      ▷ $q'$ is confident (no escalation)
10:              **if** $q'_{t+j}(v) \geq (1 - \alpha_1) M'$ **then**      ▷ $q'$ accepts $p$'s token
11:                  accept $v$      ▷ keep $\tilde{x}_{t+j}$
12:                  optionally update $\delta_1 \leftarrow$ running-avg$\big[\mathbf{1}\{q'_{t+j}(v) \geq (1 - \alpha_1)M'\}\big]$
13:              **else**
14:                  rewrite $x_{t+j} \sim q'_{t+j}$      ▷ $q'$ replaces the token
15:                  $j^* \leftarrow j$; **break**      ▷ earliest change triggers rollback
16:              **end if**
17:          **else**      ▷ $q'$ not confident $\Rightarrow$ escalate to $q$
18:              invoke $q$ on prefix $x_{<t+j}$; let $M \leftarrow \max_u q_{t+j}(u)$
19:              **if** $q$ accepts $v$ under the chosen rule (lossless or $\pi$-based) **then**
20:                  accept $v$
21:              **else**
22:                  rewrite $x_{t+j}$ using $q$ (or residual)
23:                  $j^* \leftarrow j$; **break**
24:              **end if**
25:              optionally update $\delta_2 \leftarrow$ running-avg$\big[\mathbf{1}\{q'_{t+j}(v) < (1 - \alpha_2)M'\}\big]$
26:          **end if**
27:      **end for**
28:      **if** $j^* =$ None **then**
29:          $\kappa \leftarrow \gamma$      ▷ whole block accepted
30:      **else**
31:          $\kappa \leftarrow j^*$      ▷ longest valid prefix length
32:          fallback to $q$ at position $t + \kappa$      ▷ as in Eq. (14)
33:      **end if**
34:      append the accepted prefix $x_{t:t+\kappa-1}$ to $x$;    $t \leftarrow t + \kappa$
35: **end while**
     *Note:* The analytical mixture $\pi_t^{(q')}(v)$ in Eq. (16) uses $\delta_1 = \mathbb{P}\big[q'_t(v) \geq (1 - \alpha_1) \max_u q'_t(u)\big]$ and $\delta_2 = \mathbb{P}\big[q'_t(v) < (1 - \alpha_2) \max_u q'_t(u)\big]$ (Appendix §B.2); it does not alter the control flow above.

---

## B.2. Probabilistic Derivation of $\delta_1, \delta_2$

At decoding step $t$, let the drafter distribution be $p_t(\cdot)$, the pre-verifier distribution be $q_t'(\cdot)$, and the full verifier distribution be $q_t(\cdot)$. We denote $M_t' = \max_{u \in V} q_t'(u)$. Given thresholds $0 < \alpha_2 < \alpha_1 < 1$, define $\tau_1 = 1 - \alpha_1$, $\tau_2 = 1 - \alpha_2$, with $\tau_1 > \tau_2$. We distinguish three mutually exclusive events (relative to a drafted token $v$):

$$A_t(v) := \{ q_t'(v) \geq \tau_1 M_t' \} \qquad (q' \text{ confident and } \textbf{accepts } p\text{'s token}), \qquad (\text{B.1})$$
$$B_t(v) := \{ \tau_2 M_t' \leq q_t'(v) < \tau_1 M_t' \} \qquad (q' \text{ confident but } \textbf{rewrites } \text{with } q'), \qquad (\text{B.2})$$
$$C_t(v) := \{ q_t'(v) < \tau_2 M_t' \} \qquad (q' \text{ not confident} \Rightarrow \textbf{escalate } \text{to } q). \qquad (\text{B.3})$$

Clearly $A_t(v) \cup B_t(v) \cup C_t(v) = \Omega$.

**(1) Theoretical Analysis (Full Probability Decomposition).** Define the marginal probabilities under $v \sim p_t$:

$$\delta_2 := \mathbb{P}_{v \sim p_t}[C_t(v)], \qquad (\text{B.4})$$
$$\bar{\delta}_1 := \mathbb{P}_{v \sim p_t}[B_t(v)], \qquad (\text{B.5})$$
$$\delta_p := \mathbb{P}_{v \sim p_t}[A_t(v)]. \qquad (\text{B.6})$$

Since $\delta_p + \bar{\delta}_1 + \delta_2 = 1$, the output distribution becomes

$$\pi_t(v) = \delta_p \, p_t(v) + \bar{\delta}_1 \, q_t'(v) + \delta_2 \, q_t(v). \qquad (\text{B.7})$$

Equivalently, one can express the weights via indicator expectations:

$$\delta_p = \sum_{u \in V} p_t(u) \, \mathbf{1}\{q_t'(u) \geq \tau_1 M_t'\}, \qquad (\text{B.8})$$

$$\bar{\delta}_1 = \sum_{u \in V} p_t(u) \, \mathbf{1}\{\tau_2 M_t' \leq q_t'(u) < \tau_1 M_t'\}, \qquad (\text{B.9})$$

$$\delta_2 = \sum_{u \in V} p_t(u) \, \mathbf{1}\{q_t'(u) < \tau_2 M_t'\}. \qquad (\text{B.10})$$

This yields the three-way mixture of *accept with $p$*, *rewrite with $q'$*, and *escalate to $q$*. We refer to Eqs. (B.7)-(B.10) as the **theoretical analysis version**.

**(2) Practical Version (Confidence-Gated Mixture; aligned with main text).** In practice, we use confidence gates determined by $\alpha_1, \alpha_2$ and define the mixture weights as *unconditional* probabilities under $v \sim p_t$:

$$\delta_1 := \mathbb{P}_{v \sim p_t}\big[q_t'(v) \geq (1 - \alpha_1) \max_u q_t'(u)\big], \qquad \delta_2 := \mathbb{P}_{v \sim p_t}\big[q_t'(v) < (1 - \alpha_2) \max_u q_t'(u)\big]. \qquad (\text{B.11})$$

Introduce the Bernoulli gates

$$I_{\text{conf}}(v) = \mathbf{1}\{ q_t'(v) \geq (1 - \alpha_1) \max_u q_t'(u) \}, \qquad I_{\text{esc}}(v) = \mathbf{1}\{ q_t'(v) < (1 - \alpha_2) \max_u q_t'(u) \}. \qquad (\text{B.12})$$

Conditioned on these gates, the instantaneous output at step $t$ is

$$\pi_t(v \mid I_{\text{conf}}, I_{\text{esc}}) = \big(1 - I_{\text{esc}}(v)\big)\Big(\big(1 - I_{\text{conf}}(v)\big)p_t(v) + I_{\text{conf}}(v) \, q_t'(v)\Big) + I_{\text{esc}}(v) \, q_t(v). \qquad (\text{B.13})$$

Taking expectation over $v \sim p_t$ and the gates yields the confidence-gated mixture

$$\pi_t(v) = (1 - \delta_2)\Big((1 - \delta_1) \, p_t(v) + \delta_1 \, q_t'(v)\Big) + \delta_2 \, q_t(v), \qquad (\text{B.14})$$

which matches Eq. (14) in the main text. In words, a larger $\delta_1$ (induced by a stricter $\alpha_1$) increases the weight of $q'$ within the non-escalation branch, and a smaller $\delta_2$ (from a looser $\alpha_2$) reduces escalation to $q$, stabilizing performance.

*Remark (normalized conditional alternative).* If one prefers a branch-normalized view, define

$$\widetilde{\delta}_1 \ := \ \mathbb{P}_{v \sim p_t}\big[q'_t(v) \geq (1 - \alpha_1) \max_u q'_t(u) \ \big| \ q'_t(v) \geq (1 - \alpha_2) \max_u q'_t(u)\big] \ = \ \frac{\mathbb{P}[B_t(v)]}{1 - \delta_2}, \tag{B.15}$$

and replace $\delta_1$ in Eq. (B.11) by $\widetilde{\delta}_1$; the functional form is unchanged, and both parameterizations are equivalent up to re-scaling within the non-escalation branch.

This conditional form corresponds to the implementation in Algorithm 2. We refer to Eqs. (B.11) and (B.14) as the **practical version**, which matches Eq. (14) in the main text.

**Relation to $\alpha_1, \alpha_2$.** With $\tau_1 = 1 - \alpha_1$, $\tau_2 = 1 - \alpha_2$, the three regions correspond to:

- **Acceptance region** $A_t$: $q'_t(v) \geq (1 - \alpha_1) \max_u q'_t(u)$,

- **Rewrite region** $B_t$: $(1 - \alpha_2) \max_u q'_t(u) \leq q'_t(v) < (1 - \alpha_1) \max_u q'_t(u)$,

- **Escalation region** $C_t$: $q'_t(v) < (1 - \alpha_2) \max_u q'_t(u)$.

> **Insight.** *While in the main analysis we treat $\alpha_1$ and $\alpha_2$ as **fixed hyperparameters**, one may also consider optimizing them directly during training or tuning. In this view, $\alpha_1, \alpha_2$ can be parameterized as learnable gates, with gradients obtained from the mixture distribution $\pi_t(v)$ in Eq. (B.14). Such an approach allows the system to adaptively calibrate the strictness of the confidence and escalation thresholds, potentially yielding better trade-offs between accuracy and efficiency compared to manually chosen constants.*

## B.3. Analysis of Threshold Parameters $\alpha_1, \alpha_2$

We analyze how the two thresholds control routing and, consequently, the reliance on the large model $q$. Recall that the pre-verifier $q'$ accepts a drafted token $v$ from $p$ when

$$q'_t(v) \ \geq \ (1 - \alpha_1) \max_u q'_t(u), \tag{B.16}$$

and escalation to $q$ occurs when

$$q'_t(v) \ < \ (1 - \alpha_2) \max_u q'_t(u), \tag{B.17}$$

with the token-level mixture $\pi_t^{(q')}(v) = (1 - \delta_2)\big((1 - \delta_1)p_t(v) + \delta_1 q'_t(v)\big) + \delta_2 q_t(v)$ where $\delta_1 = \Pr\big[q'_t(v) \geq (1 - \alpha_1) \max_u q'_t(u)\big]$, $\delta_2 = \Pr\big[q'_t(v) < (1 - \alpha_2) \max_u q'_t(u)\big]$.

**Monotonicity.** Define the *reliance on $q$* at step $t$ as the escalation probability $\mathcal{R}_t := \delta_2$. Because the event set $E_{\alpha_2} = \{q'_t(v) < (1 - \alpha_2) \max_u q'_t(u)\}$ shrinks as $\alpha_2$ increases (the threshold $(1 - \alpha_2)$ decreases), we have

$$\alpha_2 \uparrow \quad \Rightarrow \quad E_{\alpha_2} \downarrow \quad \Rightarrow \quad \mathcal{R}_t = \Pr(E_{\alpha_2}) \text{ is non-increasing.} \tag{B.18}$$

Thus, larger $\alpha_2$ *reduces* escalation (smaller $\delta_2$) and lowers dependence on $q$; smaller $\alpha_2$ does the opposite.

For the acceptance gate, the event $A_{\alpha_1} = \{q'_t(v) \geq (1 - \alpha_1) \max_u q'_t(u)\}$ *expands* as $\alpha_1$ increases, hence

$$\alpha_1 \uparrow \quad \Rightarrow \quad A_{\alpha_1} \uparrow \quad \Rightarrow \quad \delta_1 = \Pr(A_{\alpha_1}) \text{ is non-decreasing.} \tag{B.19}$$

Therefore, larger $\alpha_1$ makes $q'$ *less* strict (more tokens accepted by $q'$ rather than rewritten or escalated), while smaller $\alpha_1$ enforces stricter early filtering.

**Implications for Efficiency/Quality.** Let $c_p, c_{q'}, c_q$ be the per-token inference costs of $p, q', q$ with $c_p \ll c_{q'} < c_q$. Ignoring rollbacks for clarity, the expected per-token cost satisfies

$$\mathbb{E}[C_t] \approx (1 - \delta_2)\big((1 - \delta_1)c_p + \delta_1 c_{q'}\big) + \delta_2 c_q. \tag{B.20}$$

Hence,

$$\frac{\partial \mathbb{E}[C_t]}{\partial \alpha_2} = \frac{\partial \mathbb{E}[C_t]}{\partial \delta_2} \cdot \frac{\partial \delta_2}{\partial \alpha_2} \ \leq \ (c_q - ((1 - \delta_1)c_p + \delta_1 c_{q'})) \cdot 0 \ \leq \ 0, \tag{B.21}$$

since $\partial\delta_2/\partial\alpha_2 \leq 0$. Increasing $\alpha_2$ lowers cost (improves speed) by suppressing escalation to $q$; decreasing $\alpha_2$ trades speed for quality by invoking $q$ more often. Similarly,

$$\frac{\partial\mathbb{E}[C_t]}{\partial\alpha_1} = \frac{\partial\mathbb{E}[C_t]}{\partial\delta_1} \cdot \frac{\partial\delta_1}{\partial\alpha_1} = \big(c_{q'} - c_p\big) \cdot \frac{\partial\delta_1}{\partial\alpha_1} \geq 0, \tag{B.22}$$

so larger $\alpha_1$ (looser pre-verification) slightly *increases* the share handled by $q'$ (vs. $p$), raising cost mildly; smaller $\alpha_1$ (stricter) pushes more tokens away from $q'$ (either rewritten less or escalated), which can decrease cost but may increase rollback/escalation rates depending on $\alpha_2$.

## C. Comparison of Divergence Measures in $\Pi$-Space

In the main text we introduced the $\Pi$-space formulation and highlighted the role of divergence measures in connecting rejection rate with geometric discrepancy between $p$ and $q$. Here we provide a more complete comparison of several widely used distances.

**Total Variation (TV) Distance.**

$$D_{\mathrm{TV}}(p, q) \;=\; \tfrac{1}{2} \sum_{v \in V} |p(v) - q(v)|. \tag{C.1}$$

TV is symmetric and satisfies the triangle inequality. Moreover, in the lossless case ($\alpha = 0$), the rejection rate coincides exactly with $D_{\mathrm{TV}}(p, q)$. This tight consistency makes TV appealing, but also restrictive: since the direct path $p \to q$ is always shortest, introducing intermediate distributions cannot reduce the effective distance. Thus, TV provides a faithful but rigid geometry where rejection rate is tightly locked to the global discrepancy.

**Kullback–Leibler (KL) Divergence.**

$$D_{\mathrm{KL}}(p\|q) \;=\; \sum_{v \in V} p(v) \log \frac{p(v)}{q(v)}. \tag{C.2}$$

KL is asymmetric and does not satisfy the triangle inequality. Instead, it admits a non-Euclidean structure induced by negative entropy. This allows the possibility of information projection: for a convex feasible set $S$, one can find

$$r^* = \arg\min_{r \in S} D_{\mathrm{KL}}(p\|r), \tag{C.3}$$

and the generalized Pythagorean theorem guarantees

$$D_{\mathrm{KL}}(p\|q) \;\geq\; D_{\mathrm{KL}}(p\|r^*) + D_{\mathrm{KL}}(r^*\|q), \qquad \forall q \in S. \tag{C.4}$$

Hence, unlike TV, an intermediate distribution $r^*$ can strictly reduce the effective discrepancy. This property provides the theoretical foundation for reducing rejection rate by designing suitable verification pathways.

**Jensen–Shannon (JS) Divergence.**

$$D_{\mathrm{JS}}(p\|q) \;=\; \tfrac{1}{2} D_{\mathrm{KL}}\big(p\big\|\tfrac{p+q}{2}\big) + \tfrac{1}{2} D_{\mathrm{KL}}\big(q\big\|\tfrac{p+q}{2}\big). \tag{C.5}$$

JS is symmetric and bounded, and its square root defines a metric. However, due to its averaging nature, JS tends to blur directional information and does not directly correspond to rejection probabilities. Thus, while JS is useful for visualization and bounded analysis, it lacks operational meaning for speculative decoding.

**Wasserstein Distance.**

$$W(p, q) \;=\; \inf_{\gamma \in \Gamma(p,q)} \mathbb{E}_{(u,v) \sim \gamma}\big[d(u, v)\big], \tag{C.6}$$

where $\Gamma(p, q)$ is the set of couplings of $p$ and $q$, and $d(\cdot, \cdot)$ is a ground metric. Wasserstein distance captures geometry over the support of distributions and is popular in generative modeling. However, computing it is significantly more expensive, and the link to rejection rate in speculative decoding is indirect.

**Why Choose KL over TV?** To summarize:

- TV provides exact rejection consistency but forbids improvement via intermediate distributions; KL relaxes geometric constraints, allowing the introduction of an intermediate $r^*$ to lower effective discrepancy.

- This flexibility is critical in hierarchical or lossy pipelines, where rejection dynamics depend not only on global discrepancy but also on the design of intermediate verification layers.

---

**Insight.** *TV yields a faithful but rigid measure of rejection (cf. Eq. (3)), whereas KL offers a more flexible geometry that enables effective discrepancy reduction via intermediate distributions (cf. Eq. (4)). This property motivates our choice of KL as the primary divergence measure in $\Pi$-space.*

---

# D. Families of Intermediate Distributions and Minimal Piecewise KL Paths

## D.1. An Infinite Family of Beneficial Intermediates

Recall the feasible family of intermediate distributions $\mathcal{U} \subseteq \Delta_V$, which is consistent with task priors, model structure, and deployment constraints. Typical constructions include affine/moment constraints, exponential-family closures, or spans induced by deployable proxy models (*e.g.*, series variants or layer-skipped submodels). For theoretical clarity, we assume $\mathcal{U}$ to be a non-empty closed convex set.

**Lemma A.1 (Infinitely Many Beneficial Intermediates).** Fix $p, q \in \Delta_V$ and a non-singleton closed convex set $\mathcal{U}$ that contains $q$. Define the beneficial set

$$\mathcal{U}_{\text{ben}} = \left\{ u \in \mathcal{U} \;\middle|\; D_{\text{KL}}(p\|q) \geq D_{\text{KL}}(p\|u) + D_{\text{KL}}(u\|q) \right\}. \tag{D.1}$$

Then $\mathcal{U}_{\text{ben}}$ is typically infinite. In particular, let $u^* = \arg\min_{u \in \mathcal{U}} D_{\text{KL}}(p\|u)$ be the $I$-projection of $p$ onto $\mathcal{U}$. Whenever $\mathcal{U}$ contains an $I$-orthogonal submanifold through $u^*$ (or nontrivial local perturbation families around $q$), every point $u$ on such structures belongs to $\mathcal{U}_{\text{ben}}$.

*Sketch.* By the generalized Pythagorean theorem on $I$-projections, for all $q \in \mathcal{U}$,

$$D_{\text{KL}}(p\|q) \geq D_{\text{KL}}(p\|u^*) + D_{\text{KL}}(u^*\|q). \tag{D.2}$$

$u^*$ need not be unique globally if $\mathcal{U}$ admits $I$-orthogonal directions through $u^*$, and local deformations preserving $q \in \mathcal{U}$ generate continua of $u$ with the same (or smaller) two-segment cost. Hence $\mathcal{U}_{\text{ben}}$ is generically infinite. $\square$

The lemma formalizes that the theoretical improvement via a piecewise path is *not* tied to a unique $u$: there usually exist infinitely many $u \in \mathcal{U}$ that are beneficial, which motivates designing *families* of intermediates in hierarchical pipelines rather than relying on a single pivot.

## D.2. Shortest Piecewise Path via a Chain of Intermediates

The KL divergence is not a metric; thus "shortest path" needs an operational definition. We define the *piecewise KL action* of a chain as the additive cost of its segments: for a sequence $p = u_0, u_1, \ldots, u_n, u_{n+1} = q$ with $u_i \in \mathcal{U}$,

$$\mathcal{L}_{\text{KL}}(u_{0:n+1}) := \sum_{i=0}^{n} D_{\text{KL}}(u_i \,\|\, u_{i+1}). \tag{D.3}$$

Our goal is to construct chains that minimize $\mathcal{L}_{\text{KL}}$ under modeling constraints.

**Theorem A.2 (Successive $I$-Projections Yield a Minimal Chain).** Let $\mathcal{U}_0 \supseteq \mathcal{U}_1 \supseteq \cdots \supseteq \mathcal{U}_n \supseteq \{q\}$ be nested non-empty closed convex sets in $\Delta_V$. Define the successive $I$-projections

$$u_1 = \arg\min_{u \in \mathcal{U}_1} D_{\text{KL}}(u_0\|u), \quad u_2 = \arg\min_{u \in \mathcal{U}_2} D_{\text{KL}}(u_1\|u), \quad \ldots, \quad u_n = \arg\min_{u \in \mathcal{U}_n} D_{\text{KL}}(u_{n-1}\|u), \tag{D.4}$$

with $u_0 = p$ and $u_{n+1} = q$. Then the generalized Pythagorean equalities telescope to give

$$D_{\text{KL}}(p\|q) = \sum_{i=0}^{n} D_{\text{KL}}(u_i\|u_{i+1}) = \mathcal{L}_{\text{KL}}(u_{0:n+1}). \tag{D.5}$$

Moreover, among all chains that respect the same nest $\{\mathcal{U}_i\}$, the successive $I$-projection chain minimizes the piecewise action $\mathcal{L}_{\text{KL}}$.

*Sketch.* Each step satisfies $D_{\text{KL}}(u_{i-1}\|q) = D_{\text{KL}}(u_{i-1}\|u_i) + D_{\text{KL}}(u_i\|q)$ for $q \in \mathcal{U}_i$ by $I$-projection orthogonality; summing over $i$ gives Equation (D.5). Any alternative $u'_i \in \mathcal{U}_i$ increases the corresponding segment cost by the optimality of the $I$-projection, hence increases the total action. $\square$

**Corollary A.3 (Geodesic Limit).** Suppose $\{\mathcal{U}_i\}$ are chosen so that the successive $I$-projection chain lies on a dual-flat geodesic (e- or m-geodesic) between $p$ and $q$ in appropriate coordinates; let the mesh be refined so that

$\max_i D_{\mathrm{KL}}(u_i \| u_{i+1}) \to 0$ as $n \to \infty$. Then the polygonal chain converges to the corresponding straight line in the dual coordinates, while the piecewise action remains $D_{\mathrm{KL}}(p \| q)$ by Eq. (D.5).

**Discussion.** Theorem A.2 provides an operational notion of "shortest" under the additive KL action: a chain built from successive $I$-projections along nested convex families achieves the minimum cost, and the limit of a finely discretized chain aligns with a straight (geodesic) trajectory in the information-geometric sense. For hierarchical verification, the intermediates $u_1, \ldots, u_n$ correspond to tiers *e.g.*, increasingly restrictive submodels or constraints), and Eq. (D.5) explains why a well-aligned multi-tier design can realize the theoretically minimal KL action from $p$ to $q$.

### D.3. Practical Choices of $u$ in Engineering Implementation

Although $u$ can, in theory, be selected from an infinite family of intermediate distributions, in real-world system design the choice is constrained by computational cost, inference latency, hardware resources, and deployment complexity. Based on prior studies and empirical experience, the practical options for $u$ can be broadly grouped into three categories:

1. **Scale-Up of the Small Model $p$.** This approach constructs $u$ by slightly enlarging the small model $p$, for example, via LoRA, adapters, prefix-tuning, or adding a few attention or feedforward layers. The main advantages are: (i) relatively low additional inference cost due to limited model expansion, and (ii) parameter sharing with $p$, which enables rapid integration. However, the expressive power of such $u$ remains limited, and it often fails to capture the richer distributional features of $q$. As a result, its ability to reduce rejection rate is weak, especially for complex samples or long-context tasks.

2. **Using an Intermediate-Size Model of the Same Family.** Many model series (*e.g.*, 2B–9B–27B) contain natural intermediate checkpoints. Choosing such a mid-size model as $u$ is straightforward: (i) it is generally more aligned with $q$ than $p$, providing more accurate verification across a wider range of inputs; (ii) it shares the same architecture, making it easily pluggable into the hierarchical verification pipeline. The drawbacks, however, are significant: it requires loading and maintaining a separate medium-scale model, which increases memory and bandwidth demands, complicates scheduling, and reduces system throughput. Hence, despite potential gains in accuracy, the engineering burden makes this choice less practical.

3. **Skip-Layer Variant of the Large Model $q$.** A more pragmatic solution is to construct $u$ directly from $q$ by skipping certain layers or extracting lightweight sub-networks. This design offers a balance between theoretical soundness and engineering feasibility: (i) *Consistency*: since $u$ and $q$ share the same parameter space, distributional alignment is naturally preserved, avoiding model inconsistency issues; (ii) *Low overhead*: unlike introducing a standalone intermediate model, a skip-layer variant requires no additional memory footprint and only modifies inference dynamics; (iii) *Flexibility*: skipping ratios can be dynamically adjusted to trade off speed and accuracy depending on task requirements; (iv) *Stability*: empirical evidence shows that skip-layer $u$ significantly reduces rejection rate and decoding cost without sacrificing generation quality.

**Hence, we choose solution 3 as the final intermediate layer.**

# E. Why the Drafter Must Be an Independent Small Model $p$ Instead of a Layer-Skipped $q''$

A natural question in the hierarchical verification framework is: since the intermediate verifier $u$ can be constructed by layer-skipping from the large model $q$, why not also derive the drafter from $q$, namely a smaller $q''$, instead of using an independently trained small model $p$? From the perspective of distributional divergence, this design is problematic.

**Distribution Mismatch.** The drafter's role is to generate a large number of candidate tokens at very low cost, while maintaining a reasonable distributional consistency with $q$. This ensures both a low rejection rate and high throughput. If we attempt to derive $q''$ from $q$ via layer-skipping, forcing it to approximate the scale of $p$, we often encounter *distribution collapse*:

$$D_{\mathrm{KL}}(q \,\|\, q'') \;\gg\; D_{\mathrm{KL}}(q \,\|\, p), \tag{E.1}$$

meaning that the KL distance between $q$ and its truncated version $q''$ is significantly larger than that between $q$ and an independently trained small model $p$.

**Underlying Reason.** An independent small model $p$ is usually pretrained or distilled from larger models, which allows it to produce stable and smooth probability distributions over the vocabulary. In contrast, $q''$ is simply a *damaged copy* of $q$. Removing critical layers severely harms its representational capacity, leading to poor calibration of probabilities and distorted token likelihoods. Formally, if we write the layer-skipping projection as

$$q'' = \Pi_z(q), \qquad z \in \{0,1\}^L, \tag{E.2}$$

where $z_\ell = 0$ indicates the $\ell$-th layer of $q$ is removed, then as $\|z\|_0$ decreases, the divergence gap

$$\Delta D \;=\; D_{\mathrm{KL}}(q\|q'') - D_{\mathrm{KL}}(q\|p) \tag{E.3}$$

tends to grow rapidly. This reflects the fact that $q''$ suffers from systematic bias rather than well-structured compression.

**Effect on Rejection Rate.** In speculative decoding, the rejection rate at step $t$ is closely linked to distributional distance. In particular, under lossless conditions we have

$$\rho_t \;=\; D_{\mathrm{TV}}(p_t, q_t), \tag{E.4}$$

and under KL-style analysis, the rejection rate is upper bounded by

$$\rho_t \;\leq\; \sqrt{\tfrac{1}{2} D_{\mathrm{KL}}(p_t\|q_t)}. \tag{E.5}$$

Thus, if $p$ is replaced with $q''$, the bound is dominated by $D_{\mathrm{KL}}(q''\|q)$, which is substantially larger than $D_{\mathrm{KL}}(p\|q)$. This directly implies that the rejection rate with $q''$ would be unacceptably high.

**Practical Implication.** If $q''$ is used as the drafter, the large model $q$ would reject its outputs much more frequently, resulting in a drastically higher rejection rate that negates the acceleration benefit of speculative decoding. By contrast, a purpose-trained small model $p$ aligns better with $q$ in distribution space, leading to smaller KL divergence, stable rejection rates, and lower overall cost.

**Summary.** Within hierarchical verification, a layer-skipped model $q''$ is a reasonable choice for the intermediate verifier $u$, but not for the drafter. The drafter must be an independent small model $p$; only this design ensures both efficiency and smoothness of distributions, avoiding KL blow-up due to distribution collapse and maintaining the acceleration gains of speculative decoding.

## F. Alternative Margin Penalties $\phi$ and Detailed Derivations

**Setup and Unification.** Recall the acceptance inequalities (main text Eqs. (8a) and (8b)):

$$q(x) \geq (1-\alpha)\,p(x) \iff \log q(x) - \log p(x) \geq \log(1-\alpha), \tag{F.1}$$

$$p(x) \leq \tfrac{1}{\beta}\,q(x) \iff \log q(x) - \log p(x) \geq \log \beta. \tag{F.2}$$

Let $m_1 := \log(1-\alpha)$ and $m_2 := \log \beta$, and define at decoding step $t$ the logits $\ell_t^p(v) = \log p_t(v)$, $\ell_t^q(v) = \log q_t(v)$, and margins

$$z_1(v) = m_1 + \ell_t^p(v) - \ell_t^q(v), \qquad z_2(v) = m_2 + \ell_t^p(v) - \ell_t^q(v). \tag{F.3}$$

Violations correspond to $z_1(v) > 0$ (falling short of the acceptance gate) and $z_2(v) > 0$ (residual replacement pressure).

**General $\phi$-Penalized Single-Step Cost.** Let $\phi : \mathbb{R} \to \mathbb{R}_{\geq 0}$ be any convex, nondecreasing penalty ("positive-part" surrogate). A general $\phi$-style cost for one step is

$$R_{\alpha,\beta}^{\phi}(q\|p)\big|_t := \underbrace{\mathbb{E}_{v\sim p_t}\big[\phi(z_1(v))\big]}_{\text{acceptance threshold}} + \underbrace{\mathbb{E}_{v\sim q_t}\big[\phi(z_2(v))\big]}_{\text{residual replacement}}. \tag{F.4}$$

Expanding expectations yields a token-wise sum (computable when $p_t, q_t$ are available):

$$R_{\alpha,\beta}^{\phi}(q\|p)\big|_t = \sum_v p_t(v)\,\phi(z_1(v)) + \sum_v q_t(v)\,\phi(z_2(v)). \tag{F.5}$$

Accumulating over time $t \in \mathcal{T}$ gives the block-level cost

$$C_{\alpha,\beta}^{\phi}(q\|p\,|\,\pi) = \sum_{t\in\mathcal{T}} R_{\alpha,\beta}^{\phi}(q\|p)\big|_t, \qquad p_t, q_t \in \Delta_V \text{ induced by } \pi. \tag{F.6}$$

**Canonical Choices for $\phi$ (Family and Properties).** We list common surrogates, all convex and nondecreasing in $z$; gradients are useful for tuning:

1. **Hinge / ReLU:** $\phi(z) = \max\{0, z\}$.    Subgradient: $\phi'(z) \in [0,1]$, equals 1 for $z > 0$, 0 for $z < 0$.

2. **Squared hinge:** $\phi(z) = (z_+)^2$, $z_+ := \max\{0, z\}$.    Gradient: $\phi'(z) = 2z_+\mathbf{1}\{z > 0\}$.

3. **Huber-hinge (parameter $\kappa > 0$):**

$$\phi_\kappa(z) = \begin{cases} 0, & z \leq 0, \\ \dfrac{z^2}{2\kappa}, & 0 < z \leq \kappa, \\ z - \dfrac{\kappa}{2}, & z > \kappa. \end{cases} \tag{F.7}$$

   Gradient: $0$, $z/\kappa$, $1$ in the three regions.

4. **Softplus / smooth hinge (temperature $\tau > 0$):** $\phi_\tau(z) = \tau \log(1 + e^{z/\tau})$.    Gradient: $\phi_\tau'(z) = \sigma(z/\tau) \in (0,1)$ with $\sigma$ the logistic.
   *Bounds:* $z_+ \leq \phi_\tau(z) \leq z_+ + \tau \log 2$ for all $z$.

5. **Power hinge (aggressive):** $\phi_p(z) = (z_+)^p$, $p \geq 1$.    Gradient: $\phi_p'(z) = p\,(z_+)^{p-1}\mathbf{1}\{z > 0\}$.

6. **Exponential positive-part (heavy-tail):** $\phi_\tau^{\exp}(z) = \max\{0, e^{z/\tau} - 1\}$, $\tau > 0$.    Gradient for $z > 0$: $\phi_\tau'(z) = \tfrac{1}{\tau}e^{z/\tau}$; $0$ for $z \leq 0$.

Choosing larger curvature (*e.g.*, squared hinge, exponential) penalizes large violations more aggressively; smooth surrogates (softplus, Huber-hinge) provide stable gradients while remaining consistent with the hard margin in the $\tau \downarrow 0$ or $\kappa \downarrow 0$ limit.

**Ordering and Calibration.** If $\phi_1(z) \leq \phi_2(z)$ for all $z$, then $R_{\alpha,\beta}^{\phi_1}(q\|p) \leq R_{\alpha,\beta}^{\phi_2}(q\|p)$ (same for $C^\phi$). In particular, with softplus temperature $\tau$,

$$R_{\alpha,\beta}^{\text{ReLU}}(q\|p) \leq R_{\alpha,\beta}^{\text{softplus}_\tau}(q\|p) \leq R_{\alpha,\beta}^{\text{ReLU}}(q\|p) + \tau \log 2 \cdot \Big(\underbrace{\sum_v p_t(v)}_{=1} + \underbrace{\sum_v q_t(v)}_{=1}\Big) = R_{\alpha,\beta}^{\text{ReLU}}(q\|p) + 2\tau \log 2. \tag{F.8}$$

**Gradients *w.r.t.* Logits (Useful for Tuning).** Let $\ell_t^q(w) = \log q_t(w)$:

$$\frac{\partial}{\partial \ell_t^q(w)} \sum_v p_t(v) \, \phi\big(z_1(v)\big) = -\, p_t(w) \, \phi'\big(z_1(w)\big), \tag{F.9}$$

$$\frac{\partial}{\partial \ell_t^q(w)} \sum_v q_t(v) \, \phi\big(z_2(v)\big) = q_t(w) \, \phi\big(z_2(w)\big) \;-\; q_t(w) \, \mathbb{E}_{u \sim q_t}\big[\phi(z_2(u))\big] \;-\; q_t(w) \, \phi'\big(z_2(w)\big). \tag{F.10}$$

Here, we used $\partial q_t(v)/\partial \ell_t^q(w) = q_t(v) \, (\mathbf{1}\{v = w\} - q_t(w))$ and $\partial z_2(v)/\partial \ell_t^q(w) = -\mathbf{1}\{v = w\}$. Analogous expressions hold for $\ell_t^p(w)$ if $p$ is trainable.

**Single-Pass Monte Carlo via a Mixture Sampler.** If enumerating the vocabulary is infeasible, one may draw tokens from a mixture $\pi_t^\mu = \mu \, p_t + (1 - \mu) \, q_t$ and use importance weights:

$$\mathbb{E}_{v \sim p_t}\big[\phi(z_1(v))\big] = \mathbb{E}_{v \sim \pi_t^\mu}\left[ \frac{p_t(v)}{\pi_t^\mu(v)} \, \phi(z_1(v)) \right], \tag{F.11}$$

$$\mathbb{E}_{v \sim q_t}\big[\phi(z_2(v))\big] = \mathbb{E}_{v \sim \pi_t^\mu}\left[ \frac{q_t(v)}{\pi_t^\mu(v)} \, \phi(z_2(v)) \right]. \tag{F.12}$$

A simple choice is $\mu = \frac{1}{2}$ or an adaptive $\mu$ based on acceptance rates.

**Three-Tier Extension.** For the folded path $p \to u \to q$, we reuse the same $\phi$-style cost on each segment: $C_{\alpha,\beta}^\phi(u\|p\,|\,\pi)$ and $C_{\alpha,\beta}^\phi(q\|u\,|\,\pi)$. The discriminant remains

$$\Delta_{\alpha,\beta}^\phi(u\,|\,\pi) = \, C_{\alpha,\beta}^\phi(q\|p\,|\,\pi) \;-\; \Big( C_{\alpha,\beta}^\phi(u\|p\,|\,\pi) + C_{\alpha,\beta}^\phi(q\|u\,|\,\pi) \Big), \tag{F.13}$$

which reduces to Eq. (7) when $\phi = \mathrm{ReLU}$ or $\phi = \log(1 + e^z)$.

## G. From LLR Gates to Token-wise $\phi$-Costs

**Step 0: Unifying the Two Acceptance Gates as Log-Margins.** Recall the speculative-decoding acceptance conditions (Eqs. (8a) and (8b)):

$$q(x) \geq (1 - \alpha) p(x) \iff \log q(x) - \log p(x) \geq \log(1 - \alpha), \tag{G.1}$$

$$p(x) \leq \tfrac{1}{\beta} q(x) \iff \log q(x) - \log p(x) \geq \log \beta. \tag{G.2}$$

Let $m_1 := \log(1 - \alpha)$, $m_2 := \log \beta$. At decoding step $t$, define logits $\ell_t^p(v) = \log p_t(v)$, $\ell_t^q(v) = \log q_t(v)$ and the two *log-margins*

$$z_1(v) = m_1 + \ell_t^p(v) - \ell_t^q(v), \qquad z_2(v) = m_2 + \ell_t^p(v) - \ell_t^q(v). \tag{G.3}$$

Violations of the gates correspond to $z_1(v) > 0$ (falling short of the $(1 - \alpha)$ acceptance) and $z_2(v) > 0$ (pressure to replace under the $1/\beta$ bound).

**Step 1: From Hard Constraints to a Convex Positive-Part Penalty.** Let $\phi : \mathbb{R} \to \mathbb{R}_{\geq 0}$ be a convex, nondecreasing "positive-part" surrogate (for a hard threshold one may take $\phi(z) = \max\{0, z\}$; for a smooth version $\phi(z) = \log(1 + e^z)$, etc.). We measure the *severity* of violations by $\phi(z_1)$ and $\phi(z_2)$. Because the acceptance test in practice is *applied to candidates drafted from $p$*, its shortfall should be averaged under $p$; conversely, the *residual replacement* borrows mass from $q$, so its contribution is averaged under $q$. This yields the *single-step $\phi$-style cost*

$$R_{\alpha,\beta}^{\phi}(q\|p) := \mathbb{E}_{x \sim p}\left[\phi\left(\log \frac{(1-\alpha)\,p(x)}{q(x)}\right)\right] + \mathbb{E}_{x \sim q}\left[\phi\left(\log \frac{\beta\,p(x)}{q(x)}\right)\right]. \tag{G.4}$$

Taking $\phi = \mathrm{ReLU}$ or $\phi(z) = \log(1 + e^z)$ recovers Eq. (9) in the main text.

**Step 2: Making the Expectation Explicit at Step $t$.** At a fixed step $t$ on a discrete vocabulary, expectations become token-wise sums:

$$\mathbb{E}_{x \sim p_t}\left[\phi\left(\log \frac{(1-\alpha)\,p_t(x)}{q_t(x)}\right)\right] = \sum_v p_t(v)\,\phi\big(m_1 + \ell_t^p(v) - \ell_t^q(v)\big) = \sum_v p_t(v)\,\phi\big(z_1(v)\big), \tag{G.5}$$

$$\mathbb{E}_{x \sim q_t}\left[\phi\left(\log \frac{\beta\,p_t(x)}{q_t(x)}\right)\right] = \sum_v q_t(v)\,\phi\big(m_2 + \ell_t^p(v) - \ell_t^q(v)\big) = \sum_v q_t(v)\,\phi\big(z_2(v)\big). \tag{G.6}$$

Therefore the single-step cost is

$$R_{\alpha,\beta}^{\phi}(q\|p)\big|_t = \sum_v p_t(v)\,\phi\big(z_1(v)\big) + \sum_v q_t(v)\,\phi\big(z_2(v)\big). \tag{G.7}$$

**Step 3: Specializing to $\phi(z) = \max\{0, z\}$ (ReLU) Gives Eq. (11).** Choosing $\phi(z) = \mathrm{ReLU}(z) = \max\{0, z\}$ in Eq. (G.7) yields

$$R_{\alpha,\beta}^{\mathrm{KL}}(q\|p)\big|_t = \underbrace{\sum_v p_t(v)\,\mathrm{ReLU}\big(z_1(v)\big)}_{\text{acceptance threshold term}} + \underbrace{\sum_v q_t(v)\,\mathrm{ReLU}\big(z_2(v)\big)}_{\text{residual replacement term}}, \tag{G.8}$$

which is exactly Eq. (11) in the main text, with $z_1, z_2$ defined in Eq. (G.3).

**Step 4: Accumulating over Time Gives Eq. (10).** Summing the single-step costs over decoding steps $\mathcal{T}$ in the $\Pi$-space induced by the lossy rule, we obtain the block-level cost

$$C_{\alpha,\beta}^{\phi}(q\|p\,|\,\pi) = \sum_{t \in \mathcal{T}} R_{\alpha,\beta}^{\phi}(q\|p)\big|_t, \qquad p_t, q_t \in \Delta_V \text{ induced by } \pi. \tag{G.9}$$

Taking $\phi = \mathrm{ReLU}$ (or $\phi(z) = \log(1 + e^z)$) recovers Eq. (10). $\qquad\square$

**Remarks.** *(i) The first term in Eq. (G.4) is averaged under $p$ because the acceptance gate is evaluated on drafts from $p$; the second is averaged under $q$ as it quantifies the log-margin cost when $q$ replaces $p$'s proposals. (ii) Alternative convex surrogates (squared hinge, Huber-hinge, softplus) can replace ReLU without changing the derivation; only $\phi$ changes in Eq. (G.7).*

# H. Additional Tables and Figures

## H.1. All Results on T5 and Gemma

*Table 8.* Performance comparison on WebQuestions, NaturalQA, and TriviaQA benchmarks, where *Acc.*, *Rej.*, and *Speed* denote Accuracy, Rejection Rate, and Speedup, respectively.

| Model | Method | WebQuestions | | | NaturalQA | | | TriviaQA | | |
|---|---|---|---|---|---|---|---|---|---|---|
| | | Acc. | Rej. | Speed | Acc. | Rej. | Speed | Acc. | Rej. | Speed |
| Gemma2 2B→9B | Speculative Decoding | 0.27 | 0.17 | 1.50× | 0.26 | 0.27 | 1.32× | 0.50 | 0.21 | 1.55× |
| | BiLD | 0.27 | 0.15 | 1.59× | 0.26 | 0.24 | 1.67× | 0.50 | 0.19 | 1.85× |
| | Cascade SD | 0.27 | 0.16 | 1.53× | 0.26 | 0.26 | 1.43× | 0.50 | 0.20 | 1.62× |
| | Faster Cascades | **0.28** | 0.13 | 1.65× | 0.27 | 0.22 | 1.75× | 0.52 | 0.17 | 1.90× |
| | SWiFT | 0.27 | 0.15 | 1.59× | 0.26 | 0.25 | 1.61× | 0.50 | 0.18 | 1.73× |
| | CLaSP | 0.27 | 0.14 | 1.62× | 0.26 | 0.24 | 1.67× | 0.50 | 0.17 | 1.90× |
| | VIA-SD (ours) | **0.28** | **0.10** | **1.82×** | **0.28** | **0.17** | **2.10×** | **0.53** | **0.13** | **2.33×** |
| Gemma2 2B→27B | Speculative Decoding | 0.32 | 0.27 | 1.55× | 0.32 | 0.45 | 1.54× | 0.54 | 0.24 | 1.65× |
| | BiLD | 0.32 | 0.26 | 1.63× | 0.32 | 0.41 | 1.92× | 0.54 | 0.22 | 2.15× |
| | Cascade SD | 0.32 | 0.24 | 1.71× | 0.32 | 0.42 | 1.73× | 0.54 | 0.23 | 1.81× |
| | Faster Cascades | **0.33** | 0.22 | 1.81× | 0.33 | 0.40 | 2.10× | **0.56** | 0.20 | 2.30× |
| | SWiFT | 0.32 | 0.21 | 1.89× | 0.32 | 0.41 | 1.93× | 0.54 | 0.23 | 1.82× |
| | CLaSP | 0.32 | 0.20 | 1.97× | 0.32 | 0.39 | 2.21× | 0.54 | 0.22 | 1.99× |
| | VIA-SD (ours) | 0.32 | **0.14** | **2.32×** | **0.34** | **0.30** | **2.61×** | 0.55 | **0.15** | **2.50×** |
| LLaMA2 7B→13B | Speculative Decoding | 0.26 | 0.22 | 1.42× | 0.25 | 0.34 | 1.28× | 0.49 | 0.26 | 1.48× |
| | BiLD | 0.26 | 0.20 | 1.55× | 0.25 | 0.31 | 1.55× | 0.49 | 0.23 | 1.75× |
| | Cascade SD | 0.26 | 0.21 | 1.48× | 0.25 | 0.32 | 1.40× | 0.49 | 0.24 | 1.60× |
| | Faster Cascades | **0.27** | 0.17 | 1.62× | 0.26 | 0.28 | 1.68× | 0.51 | 0.20 | 1.82× |
| | SWiFT | 0.26 | 0.19 | 1.58× | 0.25 | 0.30 | 1.60× | 0.49 | 0.22 | 1.70× |
| | CLaSP | 0.26 | 0.18 | 1.65× | 0.25 | 0.29 | 1.70× | 0.49 | 0.21 | 1.85× |
| | VIA-SD (ours) | **0.27** | **0.13** | **1.88×** | **0.27** | **0.22** | **2.05×** | **0.52** | **0.16** | **2.20×** |
| LLaMA2 7B→70B | Speculative Decoding | 0.31 | 0.30 | 1.50× | 0.31 | 0.48 | 1.45× | 0.54 | 0.27 | 1.60× |
| | BiLD | 0.31 | 0.28 | 1.65× | 0.31 | 0.44 | 1.85× | 0.54 | 0.25 | 2.05× |
| | Cascade SD | 0.31 | 0.27 | 1.72× | 0.31 | 0.46 | 1.65× | 0.54 | 0.26 | 1.80× |
| | Faster Cascades | **0.32** | 0.24 | 1.85× | 0.32 | 0.42 | 2.00× | **0.56** | 0.22 | 2.15× |
| | SWiFT | 0.31 | 0.23 | 1.92× | 0.31 | 0.43 | 1.90× | 0.54 | 0.24 | 1.90× |
| | CLaSP | 0.31 | 0.22 | 2.00× | 0.31 | 0.41 | 2.10× | 0.54 | 0.23 | 2.00× |
| | VIA-SD (ours) | 0.31 | **0.16** | **2.30×** | **0.33** | **0.33** | **2.55×** | 0.55 | **0.18** | **2.45×** |
| Qwen 7B→14B | Speculative Decoding | 0.31 | 0.20 | 1.50× | 0.31 | 0.32 | 1.35× | 0.54 | 0.24 | 1.58× |
| | BiLD | 0.31 | 0.18 | 1.65× | 0.31 | 0.29 | 1.65× | 0.54 | 0.22 | 1.85× |
| | Cascade SD | 0.31 | 0.19 | 1.58× | 0.31 | 0.30 | 1.50× | 0.54 | 0.23 | 1.70× |
| | Faster Cascades | 0.32 | 0.15 | 1.72× | 0.32 | 0.26 | 1.80× | 0.56 | 0.19 | 1.95× |
| | SWiFT | 0.31 | 0.16 | 1.70× | 0.31 | 0.28 | 1.75× | 0.54 | 0.21 | 1.85× |
| | CLaSP | 0.31 | 0.15 | 1.78× | 0.31 | 0.27 | 1.85× | 0.54 | 0.20 | 2.00× |
| | VIA-SD (ours) | **0.33** | **0.11** | **2.05×** | **0.33** | **0.20** | **2.25×** | **0.57** | **0.15** | **2.40×** |
| Qwen 7B→72B | Speculative Decoding | 0.36 | 0.28 | 1.60× | 0.36 | 0.47 | 1.55× | 0.60 | 0.26 | 1.70× |
| | BiLD | 0.36 | 0.26 | 1.75× | 0.36 | 0.43 | 1.95× | 0.60 | 0.24 | 2.10× |
| | Cascade SD | 0.36 | 0.25 | 1.85× | 0.36 | 0.45 | 1.70× | 0.60 | 0.25 | 1.90× |
| | Faster Cascades | **0.37** | 0.22 | 2.00× | 0.37 | 0.41 | 2.15× | **0.62** | 0.21 | 2.25× |
| | SWiFT | 0.36 | 0.21 | 2.05× | 0.36 | 0.42 | 2.05× | 0.60 | 0.23 | 2.00× |
| | CLaSP | 0.36 | 0.20 | 2.15× | 0.36 | 0.40 | 2.25× | 0.60 | 0.22 | 2.10× |
| | VIA-SD (ours) | **0.37** | **0.15** | **2.45×** | **0.38** | **0.32** | **2.70×** | 0.61 | **0.17** | **2.55×** |

**Hyperparameters.** All the results confirm that with $\gamma = 5$, $\alpha_1 = 0.5$, and $\alpha_2 = 0.3$, the proposed approach outperforms all baselines in terms of the trade-off between quality, rejection rate, and speedup, thereby validating the effectiveness of the hierarchical verification design. This configuration strikes a balance between the acceptance tolerance of the intermediate verifier and the replacement flexibility of the final verifier, enabling stable comparisons across different decoding strategies.

**Gemma Results Analysis.** From the results in Table 8, we set temperature $T = 1$ and observe that across both **Gemma2-2B→9B** and **Gemma2-2B→27B**, our method consistently outperforms baselines on GSM8K, MBPP, SQuAD 2.0, WebQuestions, NaturalQA, and TriviaQA. The improvements can be summarized as follows.

Overall, the differences in quality scores among methods are relatively small, indicating that most techniques primarily target

*Table 9.* Performance comparison (Xsum, CNNDM, and WMT14) under different decoding settings.

| Model | Method | XSum | | | CNNDM | | | WMT14 | | |
|---|---|---|---|---|---|---|---|---|---|---|
| | | ROUGE-2 | Rej. | Speed | ROUGE-2 | Rej. | Speed | BLEU | Rej. | Speed |
| | | **Greedy Decoding: Temperature=0** | | | | | | | | |
| T5 S→L | Speculative Decoding | 16.36 | 0.33 | 1.20× | 11.00 | 0.34 | 2.05× | 18.00 | 0.39 | 1.40× |
| | BiLD | 16.36 | 0.32 | 1.25× | 11.00 | 0.33 | 2.10× | 18.00 | 0.38 | 1.55× |
| | Cascade SD | 16.36 | 0.31 | 1.30× | 11.00 | 0.32 | 2.15× | 18.00 | 0.38 | 1.55× |
| | Faster Cascades | 19.90 | 0.30 | 1.42× | 15.70 | 0.33 | 2.12× | **27.50** | 0.35 | 1.85× |
| | SWiFT | 16.36 | 0.30 | 1.35× | 11.00 | 0.31 | 2.18× | 18.00 | 0.37 | 1.70× |
| | CLaSP | 16.36 | 0.29 | 1.38× | 11.00 | 0.30 | 2.20× | 18.00 | 0.36 | 1.75× |
| | VIA-SD (ours) | **21.30** | **0.26** | **2.15×** | **12.70** | **0.28** | **2.35×** | 21.92 | **0.32** | **2.50×** |
| T5 S→XL | Speculative Decoding | 18.70 | 0.36 | 1.28× | 12.80 | 0.35 | 1.95× | 22.95 | 0.41 | 1.80× |
| | BiLD | 18.70 | 0.34 | 1.35× | 12.80 | 0.34 | 2.00× | 22.95 | 0.39 | 1.92× |
| | Cascade SD | 18.70 | 0.33 | 1.40× | 12.80 | 0.33 | 2.05× | 22.95 | 0.38 | 2.00× |
| | Faster Cascades | **18.90** | 0.31 | 1.60× | 12.95 | 0.32 | 2.10× | 23.05 | 0.31 | 2.70× |
| | SWiFT | 18.70 | 0.32 | 1.45× | 12.80 | 0.32 | 2.12× | 22.95 | 0.34 | 2.20× |
| | CLaSP | 18.70 | 0.31 | 1.48× | 12.80 | 0.31 | 2.15× | 22.95 | 0.33 | 2.25× |
| | VIA-SD (ours) | **18.90** | **0.26** | **1.95×** | **12.99** | **0.27** | **2.75×** | **23.10** | **0.27** | **3.35×** |
| | | **Non-Greedy Sampling: Temperature=1** | | | | | | | | |
| T5 S→L | Speculative Decoding | 14.80 | 0.36 | 1.02× | 11.20 | 0.36 | 1.79× | 18.10 | 0.42 | 1.30× |
| | BiLD | 14.80 | 0.34 | 1.10× | 11.20 | 0.34 | 1.85× | 18.10 | 0.41 | 1.40× |
| | Cascade SD | 14.80 | 0.33 | 1.15× | 11.20 | 0.33 | 1.90× | 18.10 | 0.40 | 1.45× |
| | Faster Cascades | 15.05 | 0.30 | 1.30× | 12.63 | 0.34 | 1.88× | **22.50** | 0.36 | 1.78× |
| | SWiFT | 14.80 | 0.32 | 1.20× | 11.20 | 0.32 | 1.92× | 18.10 | 0.38 | 1.55× |
| | CLaSP | 14.80 | 0.31 | 1.25× | 11.20 | 0.31 | 1.95× | 18.10 | 0.37 | 1.60× |
| | VIA-SD (ours) | **15.27** | **0.24** | **1.90×** | **12.81** | **0.26** | **2.10×** | 21.33 | **0.34** | **2.30×** |
| T5 S→XL | Speculative Decoding | 18.90 | 0.38 | 1.12× | 12.90 | 0.38 | 1.70× | 23.00 | 0.44 | 1.70× |
| | BiLD | 18.90 | 0.36 | 1.20× | 12.90 | 0.36 | 1.82× | 23.00 | 0.42 | 1.80× |
| | Cascade SD | 18.90 | 0.35 | 1.25× | 12.90 | 0.35 | 1.85× | 23.00 | 0.41 | 1.85× |
| | Faster Cascades | **18.99** | 0.30 | 1.45× | 12.95 | 0.34 | 1.95× | 23.05 | 0.39 | 2.55× |
| | SWiFT | 18.90 | 0.34 | 1.30× | 12.90 | 0.34 | 1.95× | 23.00 | 0.40 | 2.00× |
| | CLaSP | 18.90 | 0.33 | 1.35× | 12.90 | 0.33 | 2.00× | 23.00 | 0.39 | 2.05× |
| | VIA-SD (ours) | 18.95 | **0.28** | **1.65×** | **13.05** | **0.24** | **2.40×** | **23.10** | 0.37 | **3.10×** |

efficiency. Nevertheless, our method achieves stable and sometimes notable gains. For example, on GSM8K, the quality rises from 0.70 with the baseline Speculative Decoding to 0.73 for Gemma2-2B→9B, and further to 0.78 for Gemma2-2B→27B. Similar improvements are observed on WebQuestions and NaturalQA, showing that reducing rejection does not compromise output quality, but can even enhance it.

Rejection rate is a key indicator of the strictness and stability of verification. Compared to baseline Speculative Decoding, our method substantially lowers rejection rates across tasks and scales. For instance, on NaturalQA with Gemma2-2B→27B, rejection rate drops from 0.45 to 0.30, while on TriviaQA it decreases from 0.24 to 0.15. This indicates more effective filtering of drafted candidates, leading to fewer rollbacks.

Our method consistently achieves lower latency and higher speedup, with advantages becoming more pronounced on larger models. On TriviaQA with Gemma2-2B→27B, the speedup reaches 2.50×, clearly surpassing Faster Cascades (2.30×) and CLaSP (1.99×). On NaturalQA, the maximum speedup is 2.61×, highlighting the method's effectiveness in mitigating inference delays and improving throughput in challenging tasks. Compared to other multi-stage speculative decoding approaches (BiLD, Cascade SD, Faster Cascades, SWiFT, and CLaSP), our method achieves a well-balanced outcome across all three metrics: slight gains in quality, substantial reduction in rejection rate, and the highest speedup. This "triple-win" effect makes the approach highly practical, especially for latency-sensitive applications.

**T5 Results Analysis.** Table 9 reports the performance of T5 under different decoding strategies on XSum, CNNDM, and WMT14. For the smaller variant **T5-S→L**, our method achieves the best balance between quality, rejection rate, and speedup. On XSum, the ROUGE-2 score improves to 21.3 with a rejection rate reduced to 0.26, yielding a 2.15× acceleration. Similarly, on CNNDM we reach 12.7 ROUGE-2 with a rejection rate of 0.28, corresponding to a 2.35× speedup. On WMT14, our method obtains 21.92 BLEU and 2.50× speedup, surpassing all baselines. These results demonstrate that even for smaller-scale T5, hierarchical verification provides consistent gains without sacrificing quality.

For the larger **T5-S→XL**, the advantage becomes more evident. On XSum, our method reaches 18.9 ROUGE-2 with a rejection rate of 0.26, improving speedup to 1.95×, higher than Cascade SD (1.40×) and Faster Cascades (1.60×). On CNNDM, our approach delivers 12.99 ROUGE-2 and 2.75× speedup, while on WMT14 we obtain 23.10 BLEU and 3.35× speedup, marking the highest performance among all compared methods. The larger model shows both lower rejection rates and broader tolerance to longer drafts, thus benefiting more significantly from our hierarchical design.

Overall, these findings confirm that the proposed method scales effectively with model size. Compared to traditional speculative decoding and its variants, our approach reduces rejection by up to 0.1 absolute points and increases speedup by 0.5–1.0×, while preserving or even improving generation quality.

### H.2. Additional Ablation Studies

**Effect of $\alpha_1$ and $\alpha_2$.** Figure 6 presents the ablation results of $\alpha_1$ (strictness of rejecting the drafter $p$) and $\alpha_2$ (strictness of deferring to the full verifier $q$) on Gemma2-2B→9B. The vertical axis denotes $\alpha_1$, the horizontal axis denotes $\alpha_2$, and the color intensity reflects the relative reliance on the large model $q$. Several observations can be drawn.

First, $\alpha_2$ exerts the dominant influence. With small values of $\alpha_2$ (*e.g.*, 0.1–0.3), the system rarely escalates to the large model, keeping reliance at a low level (around 0.1–0.2). This setting improves efficiency but risks admitting tokens that deviate from the large model's distribution, potentially degrading quality. As $\alpha_2$ increases, reliance grows significantly (above 0.6), indicating that stricter deferral triggers more frequent calls to $q$, thereby stabilizing performance at the cost of reduced speedup.

Second, $\alpha_1$ has a comparatively milder effect. Increasing $\alpha_1$ makes the pre-verifier $q'$ more selective over $p$'s tokens, modestly raising the likelihood of invoking the large model. However, the vertical variation is less pronounced than the horizontal changes driven by $\alpha_2$. This suggests that $\alpha_1$ primarily tunes the early-stage filtering, while the overall reliance is chiefly governed by $\alpha_2$.

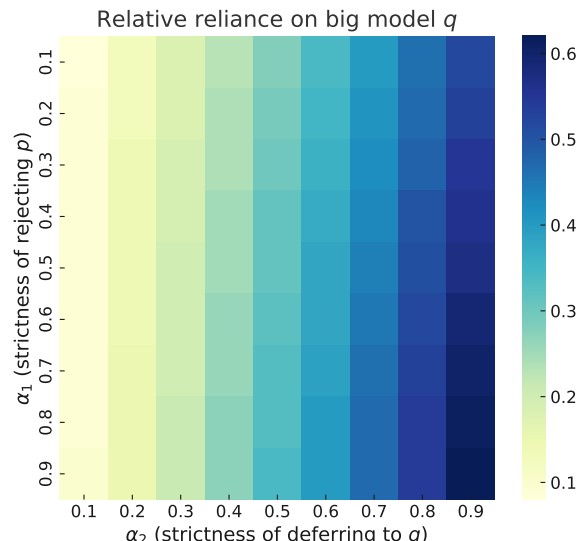

*Figure 6.* Ablation results of $\alpha_1$, $\alpha_2$ on Gemma2-2B→9B.

Finally, the joint effect of $\alpha_1$ and $\alpha_2$ reveals a meaningful trade-off space. Configurations with lower $\alpha_1$ and moderate $\alpha_2$ reduce dependence on $q$ while maintaining a reasonable balance between efficiency and quality. In contrast, setting both parameters high results in near-lossless quality but almost eliminates the computational benefits.

**Ablation on $\alpha_1$ and $\alpha_2$.** The ablation study in Figure 7 investigates the impact of varying the verification thresholds $(\alpha_1, \alpha_2)$ on the overall decoding speedup. Across all six benchmarks, we observe a consistent unimodal trend with respect to $\alpha_1$: setting $\alpha_1$ too small admits overly lenient acceptance, which increases rollback overhead, while overly strict $\alpha_1$ values cause frequent intervention by the large model. The optimal balance is typically reached around $\alpha_1 \approx 0.5$, where speculative efficiency is maximized without sacrificing verification stability.

The effect of $\alpha_2$ is similarly pronounced. Relaxed values ($\alpha_2 = 0.30$ or $0.40$) yield the best speedups across most tasks, as they reduce unnecessary escalations to the large verifier. In contrast, overly strict gating ($\alpha_2 = 0.50$) consistently degrades speedup, while overly lenient gating ($\alpha_2 = 0.20$) increases rollback, both leading to suboptimal performance. Among the datasets, SQuAD2 exhibits flatter curves, suggesting higher robustness to $\alpha_2$, whereas NaturalQA and TriviaQA are more sensitive and clearly peak around ($\alpha_1 = 0.5, \alpha_2 = 0.3$).

Overall, the ablation results highlight that the pair ($\alpha_1 = 0.5, \alpha_2 = 0.3$) provides the most stable and effective trade-off, consistently achieving the highest or near-highest speedup across all evaluated datasets.

The ablation results of $(\alpha_1, \alpha_2)$ for Gemma2 2b→27b are presented in Figure 8. Similar to the 2b→9b setting, we observe a unimodal pattern along $\alpha_1$: both overly small and overly large values reduce efficiency, while $\alpha_1 \approx 0.5$ consistently delivers the best performance across tasks.

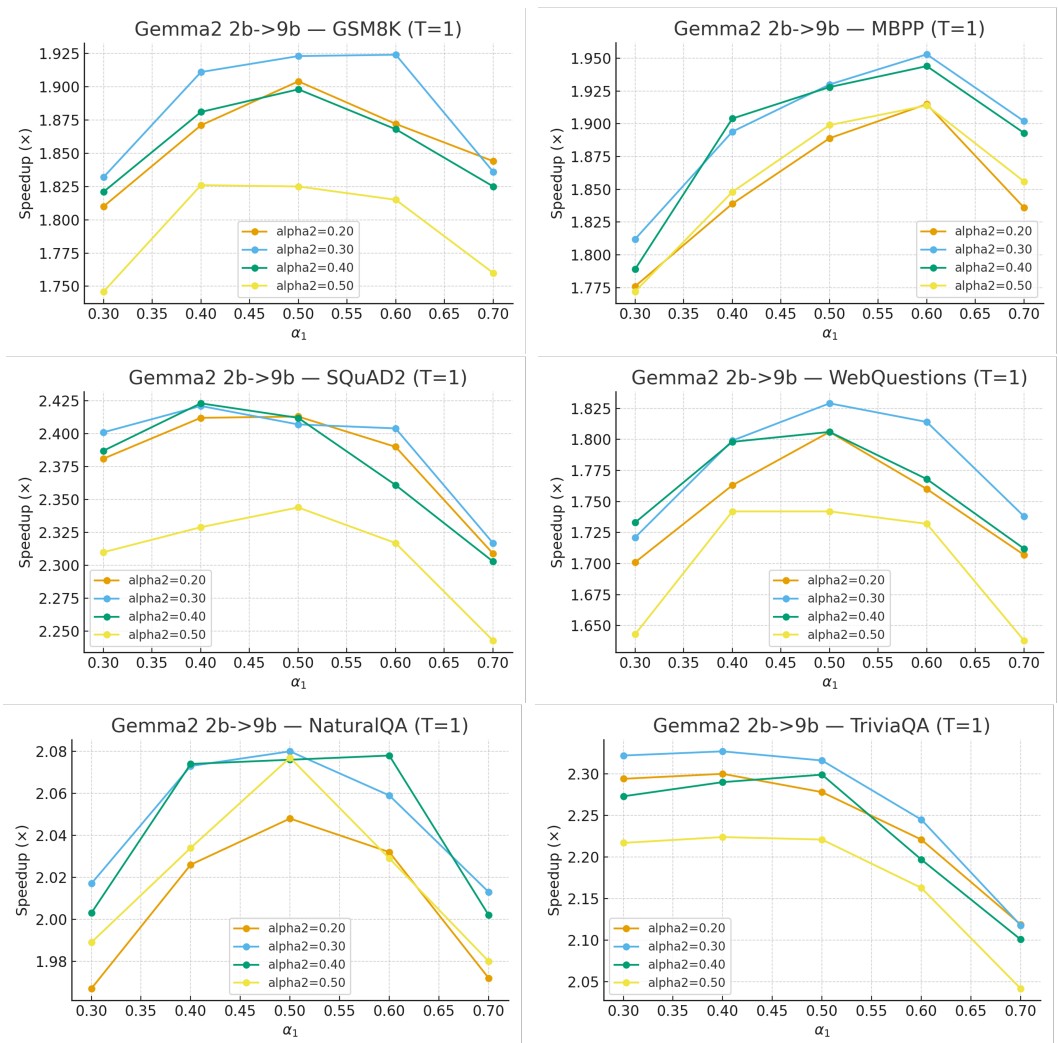

*Figure 7.* Ablation results of $\alpha_1$ and $\alpha_2$ on Gemma2-2B→9B.

For $\alpha_2$, the trade-off is more pronounced. On GSM8K and SQuAD2, relaxed settings ($\alpha_2 = 0.30$ or $0.40$) achieve the highest speedups, whereas $\alpha_2 = 0.50$ significantly suppresses acceleration due to excessive verifier intervention. MBPP exhibits a slightly right-shifted optimum, with $\alpha_1 = 0.6$ paired with $\alpha_2 = 0.30/0.40$ yielding the strongest gains. NaturalQA and TriviaQA again highlight the sensitivity to $\alpha_2$, where strict gating degrades speedup and the best region is clearly centered around ($\alpha_1 = 0.5, \alpha_2 = 0.3$).

Overall, the 2b→27b results reinforce the robustness of ($\alpha_1 = 0.5, \alpha_2 = 0.3$) as a general-purpose configuration, while also indicating slight task-specific variations such as MBPP's preference for a larger $\alpha_1$.

**Ablation on Draft Length $\gamma$.** To examine the effect of the draft length on acceleration, we vary the number of draft tokens $\gamma$ under fixed hyper-parameters ($\alpha_1 = 0.5$, $\alpha_2 = 0.3$, skip-ratio $= 45\%$). As shown in Figure 9, all datasets exhibit a consistent unimodal trend: increasing $\gamma$ initially improves speedup by enlarging the expected number of accepted tokens per draft, but overly large $\gamma$ values raise the rejection probability and verification overhead, which reduces the net gain.

For **Gemma2-2B→9B**, the optimal draft length is around $\gamma = 5$. The peak speedup reaches about $2.1\times$ on GSM8K, $2.2\times$ on MBPP, and $2.6\times$ on SQuAD2, after which the curves quickly decline. This indicates that small models are more sensitive to block rejections, as rollback costs rapidly outweigh drafting benefits when $\gamma$ becomes too large.

In contrast, **Gemma2-2B→27B** shows a broader optimum around $\gamma = 6$–$7$, where the speedup rises to approximately $2.4\times$ (GSM8K), $2.6\times$ (MBPP), and nearly $3.0\times$ (SQuAD2). Even with $\gamma = 8$–$10$, the performance remains close to the maximum, reflecting that larger models generate drafts more consistent with the verifier, thus tolerating longer blocks

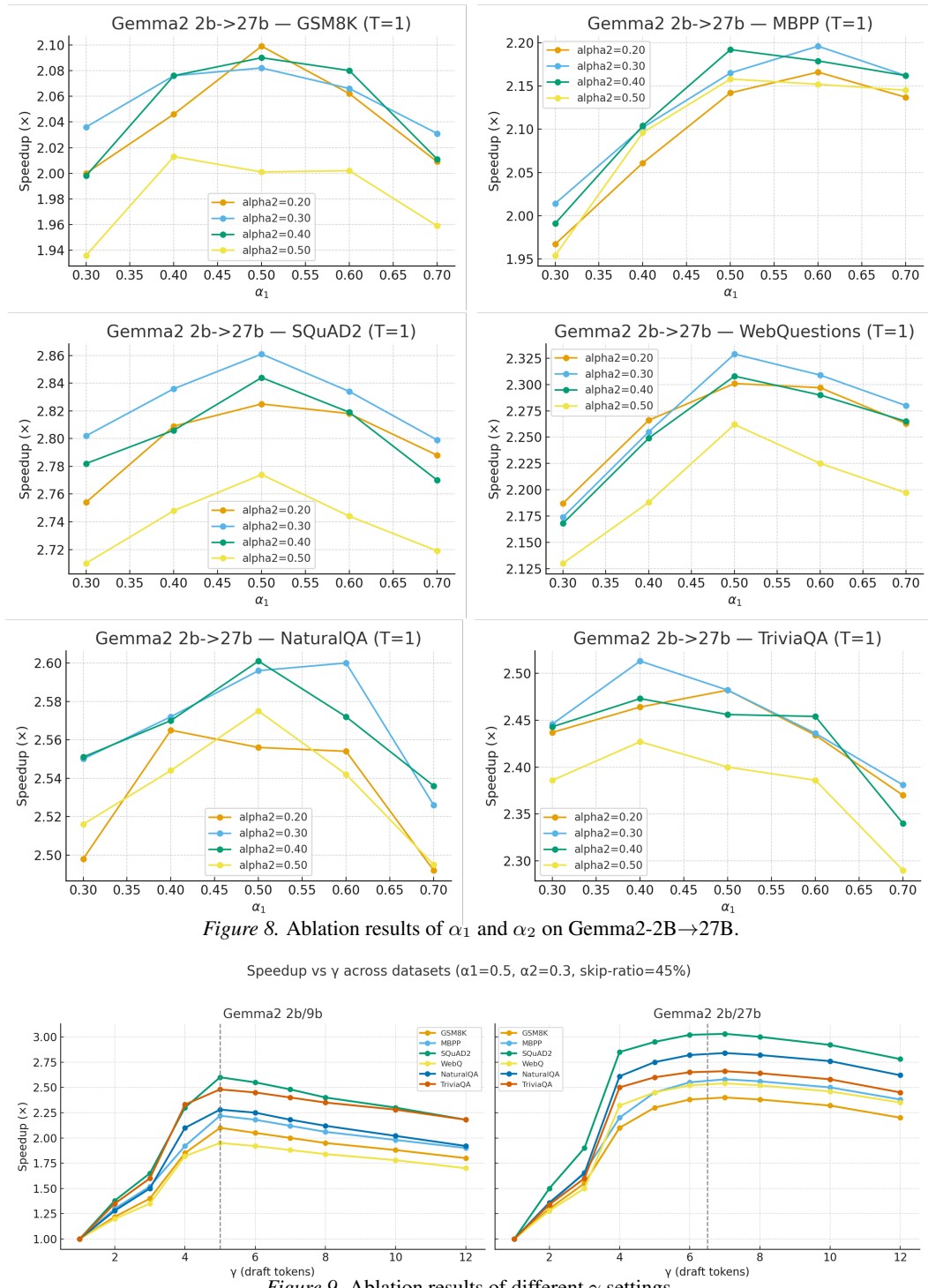

*Figure 8.* Ablation results of $\alpha_1$ and $\alpha_2$ on Gemma2-2B→27B.

*Figure 9.* Ablation results of different $\gamma$ settings.

without severe rollback penalties.

Moreover, dataset characteristics also influence the optimal $\gamma$. GSM8K, requiring long-chain reasoning, consistently yields the lowest speedup; MBPP achieves intermediate gains; while SQuAD2.0, characterized by shorter extractive outputs, attains the highest acceleration. These results suggest that $\gamma$ should be set to a moderate value ($\approx 5$ for smaller models, 6–7 for larger models) to balance draft efficiency and verification stability.

