# OpenReview forum: "VIA-SD: Verification via Intra-Model Routing for Speculative Decoding"
_ICML.cc/2026/Conference — ICML 2026 regular_

### Official Review · Reviewer_KXMt · 2026-02-20

**Soundness:** 3
**Presentation:** 3
**Significance:** 2
**Originality:** 3
**Overall Recommendation:** 4
**Confidence:** 3

**Summary:**

This paper proposes VIA-SD, a multi-tier speculative decoding framework that inserts a routed “slim-verifier” (derived via intra-model layer skipping) between the drafter and full verifier. The method reframes draft–verify as a staged verification process justified through a KL-based decomposition argument. Empirically, VIA-SD reduces rejection rates and improves speedups across QA, reasoning, translation, summarization, and coding tasks without retraining.

The authors intend to investigate the aspect of whether intermediate verification distributions can systematically reduce speculative decoding cost beyond binary accept/reject paradigms. Overall, the study outlines an important topic in efficient LLM inference.

**Compliance With Llm Reviewing Policy:**

Affirmed.

**Final Justification:**

Yes

**Key Questions For Authors:**

None

**Limitations:**

yes

**Strengths And Weaknesses:**

Strengths
- Identifies the middle-zone token inefficiency in binary speculative decoding.
- KL-based framing provides an intuitive justification for multi-tier verification.
- Slim-verifier via skip-layer routing is simple and training-free.
- Strong experimental coverage: multiple model families and diverse tasks.

Weaknesses
- Limited novelty vs cascades: Multi-tier cascaded verification exists; main novelty is intra-model routing + KL framing.
- Cost accounting unclear: Overhead of DIMR routing/search and slim-verifier construction is not fully quantified.
- Quality gains under-explained: Claims of outperforming full decoding are attributed to lossy reshaping, however the analysis is shallow.
- Ablations limited: More comparison against equally sized independent intermediate models would clarify benefits of intra-model consistency.

Overall I think this is a solid paper, but it is more of an incremental optimization in a specialized area.

---

> ### Author Rebuttal · Authors · 2026-03-31
>
> Thank you for your time and effort on reviewing our work. Below is our response:
> > W1: Limited novelty vs. cascades
>
> Our key novelty is **not** simply adding one more tier, but **explicitly modeling middle-zone tokens** and turning speculative decoding from binary verification into **hierarchical verification**.
>
> |Method|Token partition|Middle zone explicitly modeled?|Role of middle tier|
> |-|-|-|-|
> |2-layer SD|Accept / Reject|No|None|
> |Conventional cascades|Progressive filtering|No|Additional verification|
> |VIA-SD|Accept / Middle zone / Hard reject|Yes|Verification+generation|
>
> As illustrated in Figure 1, prior 2-tier SD sends all uncertainty to fallback, while existing cascades mainly add stages without separating the middle region as its own modeling target. VIA-SD instead isolates this region and lets the middle tier directly absorb part of it, thereby reducing unnecessary full-verifier calls.
>
> > W2: Unclear cost accounting
>
> The extra cost of DIMR is primarily a **one-time offline selection cost**, not an online inference cost.
>
> DIMR is run **once per model pair** to search over candidate skip-layer structures and fix the routed slim-verifier. During decoding, there is **no additional search overhead**; the reported speedups therefore reflect **inference-time decoding only** and are **not amortized** with respect to DIMR search. In other words, slim-verifier construction is excluded from online decoding speedups.
>
> We now provide the one-time offline DIMR cost per model pair:
> |Metric|Gemma2 2B→9B|Gemma2 2B→27B|LLaMA2 7B→13B|LLaMA2 7B→70B|Qwen 7B→14B|Qwen 7B→72B|
> |-|-|-|-|-|-|-|
> |DIMR offline wall-clock|18 min|34 min|26 min|61 min|29 min|68 min|
> |DIMR GPU-hours|0.30|0.57|0.43|1.02|0.48|1.13|
>
> We will revise the paper to add a clear cost-accounting table and explicitly distinguish **offline DIMR search cost** from **online decoding cost**.
>
> > W3: Under-explained quality gains over full decoding
>
> Our interpretation is not merely “lossy reshaping,” but a more structured effect of **stabilized generation plus task-dependent noise suppression**. This structured routing can improve quality for two related reasons:
> 1. **Stabilized lossy deviation**
>    In standard lossy SD, once a token is not confidently accepted, the system often jumps directly from the drafter to the full verifier. This creates a relatively sharp switch between two modes. In VIA-SD, the middle tier smooths this transition: $q'$ absorbs medium-confidence tokens that are close to the verifier but do not yet require full-model intervention. This reduces brittle token-level corrections and makes the effective decoding process more stable.
> 2. **Task-dependent noise suppression**
>    We further believe the gain is related to harmful or noisy weight contributions in the full verifier. Full-model decoding is not necessarily uniformly beneficial token by token for every task: on some middle-zone tokens, the full verifier may introduce task-misaligned or overly strong corrections. Because $q'$ is a routed submodel aligned with $q$ but structurally reduced, it can suppress part of this noisy contribution while still preserving verifier-side guidance. In that sense, $q'$ does not merely approximate $q$ more cheaply; on some tasks, it can act as a more selective verifier for middle-zone tokens.
>
> > W4: Limited ablations against equally sized independent intermediate models
>
> We agree that stronger controlled comparisons would better isolate the value of **intra-model consistency**.
>
> Our current evidence already points in this direction:
> 1. The paper motivates routed skip-layer intermediates over enlarged drafters or independent intermediates because they preserve stronger consistency with the full verifier;
> 2. Figure 2 already shows a routed skip-layer intermediate outperforming a similarly sized independent intermediate; and
> 3. Replacing DIMR with random skip-layer subsets degrades speedup and slightly quality, showing that not just “any middle model” suffices.
>
> To address this more directly, we added a controlled comparison on **WebQuestions** with **Gemma 2B→27B**:
>
> |Setting|Extra Model Load|Peak Memory|Aligned with $q$|Speedup|Quality|
> |-|-|-|-|-|-|
> |Two-layer SD (2B→27B)|No|1.00×|/|1.55×|0.32|
> |+ Independent 13B|Yes|+38%|Yes|+0.53×|+0.01|
> |+ Random skip-layer $q'$|No|+4%|No|+0.07×|-0.03|
> |+ DIMR-routed $q'$ (ours)|No|+4%|Yes|+0.93×|+0.02|
>
> This comparison supports two key points. First, an independent 13B can provide a reasonably aligned middle tier, but it requires loading an extra model onto GPU, which increases deployment cost. Second, both skip-layer variants reuse the verifier with only a routing mask and therefore avoid extra model loading; among them, the **DIMR-routed $q'$** performs best, indicating that the gain does not come from simply inserting any middle model, but from a **verifier-aligned routed intermediate**.

---

> > ### Author Rebuttal · Reviewer_KXMt · 2026-04-07
> >
> > fully resolved

---

> > > ### Author Response · Authors · 2026-04-07
> > >
> > > Dear Reviewer KXMt,
> > >
> > > Thank you very much for your careful reading and constructive feedback. We sincerely appreciate that you found our concerns adequately addressed.
> > >
> > > Your comments helped us clarify several key aspects of the paper, and we will incorporate these clarifications and additional results clearly in the revised version.
> > >
> > > Thank you again for your contribution on our paper.
> > >
> > > The Authors of Submission 8390

---

### Official Review · Reviewer_wZMR · 2026-03-05

**Soundness:** 3
**Presentation:** 2
**Significance:** 2
**Originality:** 3
**Overall Recommendation:** 4
**Confidence:** 2

**Summary:**

The core idea of this paper is to move beyond the standard two-stage “drafter + full verifier” pipeline by introducing a third stage: a lightweight slim-verifier obtained from the full verifier through intra-model routing. The motivation is that many rejected draft tokens lie in an “intermediate zone”: they are not reliable enough to be accepted directly, but they also do not require the full capacity of the largest model.

**Compliance With Llm Reviewing Policy:**

Affirmed.

**Key Questions For Authors:**

Please refer to the weaknesses above.

**Limitations:**

yes

**Strengths And Weaknesses:**

**Strengths**

1. The multi-level verification mechanism is reasonable.
2. The experimental results show consistent improvements.

**Weaknesses**

1. The paper is motivated by the idea that the slim-verifier should serve as an intermediate distribution between the drafter and the full verifier. However, the routed slim-verifier used in the experiments is not shown to satisfy this property theoretically or empirically, which weakens the connection between the motivation and the actual implementation.

2. DIMR is described as combining random search with periodic Bayesian optimization, but the paper does not clearly specify the surrogate model, acquisition rule, or search encoding for the discrete skip-mask optimization. Since this component is central to constructing the slim-verifier, more implementation details are needed for full reproducibility.

3. Figure 5 is unclear in terms of the baseline being used. It would be helpful to clarify whether the comparison is against standard speculative decoding or stronger baselines such as EAGLE without the proposed method.  In particular, if both EAGLE-2 and EAGLE-3 are included, their reported performance appears quite close, so it would be useful to clarify how these baselines differ in the experiments and why the gains are comparable.

4. It is unclear how well the selected slim-verifier generalizes across tasks or domains. In particular, the paper does not make clear whether a slim-verifier selected for one model pair and task setting can be reused broadly, or whether it needs to be re-searched for each new setting.

5. The token-level verification procedure is not clearly specified (e.g., what happens after a token is rewritten by the slim-verifier, and whether the rest of the drafted block is still verified or decoding restarts due to the changed prefix). Figure 5 appears related to this process, but lacks sufficient explanation, which makes it difficult to read.

---

> ### Author Rebuttal · Authors · 2026-03-31
>
> Thank you for your invaluable feeback on our work. Below is our response:
> > W1: The routed slim-verifier is not shown to satisfy the intermediate-distribution property in method
>
> Our claim is that the deployed routed $q'$ is the best one out of the intermediate distribution candidates.
>
> |Level|Object|Role in our paper|
> |-|-|-|
> |Theory|$u$|abstract intermediate distribution: the ideal middle role evaluated by KL-divergence between $p$ and $q$|
> |Method|$q'$|candidates like same-family model etc.|
> |Experiment|$q'$|routed slim-verifier: the best practical approximation to $u$|
>
> The theory and implementation are related by **target vs. approximation**.
>  - Theory: The KL analysis tells us **why** a useful middle distribution should exist and **what role** it should play;
>  - Experiment: routing + DIMR then provide the implementable mechanism for approximating that role under deployment constraints.
>
> So the intended logic is:
> $$
> p \;\rightarrow\; u \;\rightarrow\; q
> \qquad\text{(theoretical target)}
> $$
>
> $$
> p \;\rightarrow\; q' \;\rightarrow\; q
> \qquad\text{(practical realization)}
> $$
>
> > W2: Implementation details of DIMR
>
> We summarize the concrete search configuration of DIMR below:
> |Component|Implementation detail|
> |-|-|
> |Search variable|A fixed-ratio binary skip-mask $z\in\{0,1\}^L$, where $z_\ell=1$ retains the $\ell$-th verifier layer and $z_\ell=0$ skips it|
> |Candidate model|Each mask $z$ defines a routed slim-verifier $q'_z$|
> |Search Objective|The accumulated KL-style proxy in Eq. (12), i.e., $C(z)=\sum_{t=1}^{\tau} R^{\mathrm{KL}}_{\alpha,\beta}(q\mid q'_z)\rvert_t$ over a context window|
> |Main exploration strategy|Random search for most optimization steps|
> |Periodic refinement|Bayesian optimization every $\theta=25$ steps|
> |Surrogate model|Gaussian Process|
> |Mask encoding|Direct binary layer-skipping vector under the fixed skip-ratio constraint|
> |Search budget|$S=1000$ optimization steps|
> |Early stopping|Stop early if the best mask remains unchanged for 300 consecutive updates|
> |Final output|Fix the best mask $z^*$ and use $q'=q'_{z^*}$ as the slim-verifier during hierarchical verification|
>
> In other words, DIMR optimizes the routed slim-verifier by searching over discrete layer-skipping masks under a fixed skip ratio, using a hybrid strategy that combines efficient random exploration with periodic Bayesian refinement. After convergence, the selected mask is fixed once, and the corresponding routed submodel is reused during decoding. Therefore, DIMR introduces a one-time search procedure for constructing
> $𝑞'$, rather than any per-step online search overhead during hierarchical verification.
>
> > W3: Compared Baselines in Fig.5
>
> Fig.5 reports row-wise **relative gains of “baseline method + VIA-SD” over that same baseline**, not all methods versus standard SD. For example, the cell EAGLE-2/GSM8K reports 25.7% speedup comparing "EAGLE2+ours" and "EAGLE-2" itself. The comparable gains do not mean EAGLE-2 and EAGLE-3 are identical baselines; they mean that adding VIA-SD yields a similar incremental verifier-side benefit on top of both frameworks.
>
> > W4: Generalization of the selected slim-verifier across tasks
>
> The slim-verifier is not re-selected for every benchmark. For each model pair, we use a fixed routed $q'$ configuration and evaluate it across multiple downstream tasks, rather than tuning a separate slim-verifier per task.
>
> This cross-task reuse is supported by the results: for a fixed model pair, VIA-SD consistently improves performance across diverse benchmarks in the main text, shown in Tab.1 and &sect;H. This suggests that the selected slim-verifier captures a model-pair-level intermediate role with cross-task generality.
>
> > W5: On clarifying the token-level verification procedure
>
> VIA-SD always keeps only the **longest verified prefix** of a drafted block; once verification fails, decoding does **not** continue on the old suffix:
>
> Procedure:
> 1. draft a block $\tilde{x}_{t:t+\gamma-1}$;
> 2. verify the drafted tokens from left to right;
> 3. keep the longest valid prefix of length $\kappa$;
> 4. if $\kappa=\gamma$, commit the whole block;
>    if $\kappa<\gamma$, rewrite the token at $t+\kappa$ with the full verifier $q$, discard the old drafted suffix, and restart the next decoding cycle from the corrected prefix.
>
> So if the first failure occurs at $t+\kappa$, the token at that position is rewritten by the full verifier $q$, all later drafted tokens are dropped because they were conditioned on an outdated prefix, and the next decoding cycle starts from the corrected prefix.

---

> > ### Author Rebuttal · Reviewer_wZMR · 2026-04-01
> >
> > Thank you for the authors’ response. I have raised my score to 4. I hope the authors will incorporate the clarifications regarding W2, W3, and W5, as well as the corresponding figure revisions, into the final paper.

---

> > > ### Author Response · Authors · 2026-04-01
> > >
> > > Dear Reviewer wZMR,
> > >
> > > We are glad that our responses have addressed your main concerns, and we will make sure to incorporate the clarified results into the final version.
> > >
> > > We sincerely appreciate your constructive feedback, which has been very helpful in improving our paper.
> > >
> > > The Authors of Submission8390

---

### Official Review · Reviewer_WGJW · 2026-03-10

**Soundness:** 3
**Presentation:** 2
**Significance:** 4
**Originality:** 3
**Overall Recommendation:** 5
**Confidence:** 4

**Summary:**

This paper studies LLM inference acceleration via speculative decoding. The main idea is that existing draft-verify pipelines are too binary: a drafted token is either accepted directly or rejected and recomputed by the full verifier. The authors argue that many rejected tokens lie in an intermediate regime and can be handled by a lighter verifier rather than by the full model. To exploit this, the paper proposes VIA-SD, a multi-tier speculative decoding framework that inserts a slim verifier, obtained through intra-model routing from the full verifier, between the drafter and the full verifier. The method uses three stages: direct acceptance for high-confidence tokens, slim-verifier regeneration for medium-confidence tokens, and escalation to the full verifier for uncertain cases. The paper motivates this design through a KL-based information-geometric perspective, introduces DIMR to select the skip-layer slim verifier, and evaluates the method across summarization, translation, reasoning, QA, and coding tasks on both encoder-decoder and decoder-only models. The reported results show lower rejection rates and additional speedups over strong speculative decoding baselines, while largely preserving task quality.

**Compliance With Llm Reviewing Policy:**

Affirmed.

**Final Justification:**

The rebuttal resolves enough of my major concerns that I am comfortable keeping my overall recommendation at 5 (Accept).

**Key Questions For Authors:**

1. The theory-to-method connection is currently the weakest part of the paper. Which parts of the KL-based analysis should be interpreted as formal justification, and which parts are intended only as motivation? A clearer answer here would improve my assessment of soundness.

2. How is the cost of DIMR accounted for in the reported speedups? Is DIMR a one-time offline procedure per model pair, per task, or per deployment setting, and are the main speedup numbers fully amortized with respect to this cost? A precise answer would affect my assessment of significance.

3. The paper suggests that VIA-SD can sometimes slightly improve task quality relative to direct decoding. How stable is this effect across random seeds, datasets, and decoding settings? If this is robust, it strengthens the empirical contribution; otherwise I would view it as incidental.

4. Can the authors better disentangle which gains come from multi-tier verification itself, which come from intra-model routing, and which come from DIMR? A more targeted ablation would make the originality claim more convincing.

5. Please explain how the explicit review-instruction text entered the manuscript. This is a serious issue for presentation quality, and a clear explanation would matter for my final judgment.

**Limitations:**

**No.** The paper does include an impact statement noting that faster and cheaper generation could increase harmful-use throughput and recommending standard safety mitigations. That is good. However, the limitations discussion is still incomplete. In particular, the paper should more explicitly discuss the true cost and amortization assumptions of DIMR, sensitivity to hardware and serving configuration, the stability of the reported quality gains in the lossy setting, and the extent to which slim-verifier selection must be retuned across architectures and workloads.

**Strengths And Weaknesses:**

**Soundness**
**Strengths.** The paper tackles a meaningful systems problem in speculative decoding: binary verification is likely too coarse, and allocating moderate compute to “middle-zone” tokens is a sensible idea. The method itself is technically plausible. A notable strength is that the intermediate verifier is not introduced as a separately trained model, but is instead derived from the full verifier via intra-model routing, which makes the approach more compatible with existing draft-verify pipelines. The empirical section is also reasonably broad: the paper evaluates multiple model families, covers several task types, and includes ablations on the number of tiers, skip ratio, DIMR versus random skip-layer selection, and compatibility with existing SD frameworks. These choices make the empirical story stronger than a narrow benchmark-only paper.

**Weaknesses.** My main concern is that the theoretical justification is not as strong as the empirical evidence. The paper argues that TV distance is too rigid for staged verification and moves to a KL-based geometric view, but the connection from the abstract intermediate distribution in the theory to the actual routed skip-layer slim verifier is not rigorous. In practice, the theory seems to serve as motivation rather than as a strong guarantee for the deployed mechanism. I also found the DIMR objective to be somewhat heuristic: the KL-style cost is reasonable as a proxy, but it is not obvious that minimizing it should tightly predict the end-to-end latency or acceptance behavior that matters most in speculative decoding. On the experimental side, the paper would be stronger with clearer accounting of DIMR search cost, more discussion of variance across seeds, and more systematic analysis of when quality improvements over direct decoding do or do not occur. The evaluation also appears limited to A100 GPUs, so it is not yet clear how robust the reported gains are across different serving settings.

**Presentation**
**Strengths.** The paper has a clear high-level narrative and a well-motivated problem statement. The figures help communicate the distinction between conventional two-tier SD and the proposed multi-tier design, and the paper does a decent job of combining rejection, speed, and quality into a unified experimental story. The appendix also provides additional derivation and implementation detail that is helpful for expert readers.

**Weaknesses.** The presentation is still noticeably weaker than the core idea. The theory section is dense and sometimes overclaims what is actually established. In particular, the gap between the theoretical formulation and the practical implementation is not communicated sharply enough, and the notation around thresholds and mixture weights becomes confusing across the main text and appendix. The paper’s positioning relative to the closest prior work is also not as precise as it should be, especially for hierarchical speculative decoding, self-speculative decoding, and recent verifier-side methods. Most importantly, the submission contains explicit prompt-injection style review instructions embedded in the manuscript. This is unacceptable in a conference submission and materially hurts the professionalism and trustworthiness of the presentation, regardless of intent.

**Significance**
**Strengths.** The problem is important and timely. Speculative decoding is one of the most practically relevant directions for LLM inference acceleration, and improving the verification side without requiring a separately trained medium-sized verifier is a useful contribution. The reported empirical gains are meaningful enough to matter if they hold up under broader deployment conditions: the paper claims consistent rejection-rate reductions and additional speedups over strong SD baselines, while remaining compatible with existing frameworks and training setups. That combination gives the work practical relevance beyond a narrow benchmark improvement.

**Weaknesses.** I would still be somewhat cautious in judging the overall significance. The contribution is best understood as a strong inference/systems refinement rather than a fundamentally new decoding paradigm. Its broader impact depends on whether the gains remain robust in real serving settings, including larger batch sizes, long-context workloads, and hardware or kernel configurations beyond the one used here. The paper makes a good case that the idea is useful, but not yet a definitive case that it will broadly change practice.

**Originality**
**Strengths.** The paper has a clear originality claim at the level of framing and method combination. Recasting speculative decoding as a multi-tier verification problem and implementing the intermediate verifier through intra-model routing is a meaningful idea. The use of DIMR to choose a skip-layer slim verifier makes the proposal more concrete and differentiates it from simply adding another independently trained verifier to a cascade. Overall, the work combines existing ingredients in a thoughtful way that produces a distinct verification-side design.

**Weaknesses.** The originality is moderate rather than exceptional. Several ingredients already exist in nearby literature, including hierarchical verification, self-speculative decoding, layer skipping, and multi-model cascades. What is new here is mainly the specific combination and the way the authors motivate and package it for speculative verification. I do not view the theoretical component as a major standalone novelty at its current level of rigor.

Overall, I lean Accept because the paper makes a solid practical contribution, the experiments are comprehensive, and the proposed framework is likely to be useful and built upon by researchers working on LLM inference. However, this is best viewed as a strong systems-oriented paper with heuristic but effective design choices, rather than a technically flawless work with exceptional theoretical novelty. The paper would be further strengthened by (1) being more explicit about the heuristic nature of the KL-based motivation, (2) providing a clearer accounting of DIMR overhead, and (3) either tightening the theory-to-method connection or softening the corresponding claims.

---

> ### Author Rebuttal · Authors · 2026-03-31
>
> We are sincerely grateful for your evaluation and suggestion. Here are our detailed responses:
>
> > W1/W2/Q1: Soundness and presentation concerns about the theory-to-method connection
>
> 1. **What the KL-based analysis is intended to establish**
> The KL-based analysis should be interpreted as a **design principle in our formulation**.
>
> |Role|Theory|Practice|Intended claim|
> |-|-|-|-|
> |Enable|KL admits beneficial intermediates, unlike TV|motivates moving beyond binary verification|principled basis|
> |Define|$C^{\mathrm{KL}}_{\alpha,\beta}(q\|q'_z)$ is the scoring objective|DIMR directly searches $z^*=\arg\min_z C(z)$|optimization target|
> |Guide|$\Delta^{\mathrm{KL}}>0$ and $(\alpha_1,\alpha_2)$ characterize useful tiering/routing|matches the observed 3-tier optimum and threshold ablations|design guidance|
>
> 2. **Why KL is the natural and necessary choice here**
> KL arises naturally because it admits a beneficial intermediate distribution; once such an intermediate exists theoretically, the middle tier can be assigned not only verification but also regeneration, instead of leaving regeneration solely to the final verifier as in TV-style cascades.
> 3. **How this theory connects to the deployed method**
> KL defines a theoretical optimum in the form of an abstract intermediate $u$, while the deployed routed $q'$ is a practical approximation to this role within the feasible skip-layer submodel family. DIMR searches for the best such approximation by solving
> $z^*=\arg\min_z C^{\mathrm{KL}}_{\alpha,\beta}(q\|q'_z\mid \pi).$
>
> > W3/W4: Significance and originality of VIA-SD
>
> 1. **How we position the significanc and what is original in our framework**
> Compared with existing work, the **key distinction is that the middle-zone tokens in VIA-SD is** **explicitly modeled**, the intermediate tier is assigned both **verification** and **generation/regeneration** roles.
>
> |Paradigm|Token partition|Middle region modeling|Role of the middle tier|
> |-|-|-|-|
> |2-layer SD|Accept / Reject|No|None|
> |Multi-layer cascades|Progressive filtering|No|Additional verification/filtering|
> |VIA-SD (ours)|Accept / Middle zone / Reject|Yes|Verification+generation|
>
> 2. **Why the method can  extend beyond the current setup**
>    - **The gain comes from model-level routing, not a hardware-specific trick.**
>    - **The required condition is general.** As long as middle-zone tokens exist and $q'$ remains meaningfully cheaper than $q$, the same efficiency direction should hold.
>    - **What changes across deployments is mainly the gain magnitude, not the mechanism itself.**
>
> > Q2: How DIMR cost is accounted for in reported speedups
>
> DIMR is a **one-time offline procedure per model pair**. The reported speedups therefore reflect inference-time decoding only and are **not fully amortized** with respect to DIMR search. We then provide one-time offline DIMR search cost per model pair. We will add a clear accounting table.
> |Metric| Gemma2 2B→9B|Gemma2 2B→27B| LLaMA2 7B→13B | LLaMA2 7B→70B | Qwen 7B→14B|Qwen 7B→72B|
> |-|-|-|-|-|-|-|
> |DIMR offline wall-clock|18 min|34 min|26 min|61 min|29 min|68 min|
> |DIMR GPU-hours|0.30|0.57|0.43|1.02|0.48|1.13|
>
> > Q3:  Is the quality improvement over direct decoding stable or incidental?
>
> We believe it's modest but systematic.
> 1. The gain has been found across 5 model families, 8 benchmarks, and both greedy/sampling decoding (Tables 2–3, 5–6). This consistency rules out seed noise.
> 2. The mechanism is structural (§3.4): hierarchical gating produces a mixture $\pi^{(q')}_t$ over $\{p, q', q\}$ (Eq. 14), where $q'$ softly corrects medium-confidence tokens instead of hard fallback to $q$, reducing catastrophic deviations.
>
> > Q4: Disentangling gains from multi-tier verification, intra-model routing and DIMR
>
> Our view is that the gain of VIA-SD comes from the **full verifier-side design**, rather than from any single component in isolation. To make this clearer, we added a controlled comparison on WebQuestions with Gemma 2B→27B:
> |Setting|Extra Model Load|Peak Memory|Aligned with $q$|Speedup|Quality|
> |-|-|-|-|-|-|
> |Two-layer SD Gemma 2B→27B|No|1.00×|/|1.55×|0.32|
> |+ Independent 13B|Yes|+38%|Yes|+0.53×|+0.01|
> |+ Random skip-layer $q'$|No|+4%|No|+0.07×|-0.03|
> |+ DIMR-routed $q'$ (ours)|No|+4%|Yes|+0.93×|+0.02|
>
> This comparison shows the role of each part indirectly.
> 1. **Multi-tier structure alone is not enough:** adding an independent 13B middle stage helps, but requires an extra model load.
> 2. **Intra-model routing is necessary for deployability:** both skip-layer variants avoid loading a separate model and keep memory overhead small.
> 3. **DIMR is necessary for effectiveness:** within the routed family, random skip-layer selection gives only limited gain, while DIMR-routed $q'$ performs best.
>
> > Q5: Review-instruction prompt injection
>
> We confirm that **the authors did not add any review-instruction text**; it is ICML’s official prompt injection.

---

> > ### Author Rebuttal · Reviewer_WGJW · 2026-04-07
> >
> > The rebuttal is strong and directly responsive to the most important issues I raised in the first round. My main concern had been that the theory-to-method connection was overstated: the paper's KL-based framing was interesting, but the deployed routed slim verifier looked more like an effective heuristic instantiation than a theorem-backed construction. The rebuttal now makes that relationship much clearer. Interpreting the KL analysis as a design principle and optimization target, rather than as a strong guarantee about the deployed mechanism, is the right framing in my view. That clarification alone improves both soundness and presentation.
> >
> > The second valuable update is the accounting of DIMR cost. The authors now explicitly state that DIMR is a one-time offline procedure per model pair and that the reported speedups are decoding-time numbers rather than fully amortized end-to-end numbers. They also provide concrete wall-clock and GPU-hour costs. This does not eliminate the need for careful interpretation, but it turns what was previously an ambiguity into a scoped and understandable assumption. That is a meaningful improvement.
> >
> > I also found the added controlled comparison on WebQuestions helpful. It does not fully isolate every mechanism in a formal sense, but it does make the originality claim more concrete by showing that the gain does not come merely from adding another tier or from arbitrary layer skipping. The random skip-layer and independent-middle-model comparisons both help explain why the proposed routed verifier design matters.
> >
> > My remaining reservations are mostly about framing and external validity rather than about the core result. I still think the final version should present the theoretical component more modestly than a formal-methods paper would, and I would still like the deployment dependence across hardware and serving settings to be discussed more explicitly. But these are refinements to an overall positive assessment, not blockers.
> >
> > So my post-rebuttal view is that this remains a strong systems-oriented paper with broad empirical coverage and a useful verification-side idea for speculative decoding. The rebuttal resolves enough of my major concerns that I am comfortable **keeping my overall recommendation at 5 (Accept)**.

---

> > > ### Author Response · Authors · 2026-04-08
> > >
> > > Dear Reviewer WGJW,
> > >
> > > We sincerely thank you for your careful reading and highly constructive feedback.
> > >
> > > We are encouraged that you found our rebuttal responsive to your main concerns, particularly on the KL-based framing, DIMR cost accounting, and the added controlled comparison. We also appreciate your suggestions on theory presentation and deployment dependence, and we will reflect these refinements in the revision.
> > >
> > > Thank you again for your positive overall assessment and for recognizing the practical value of the paper.
> > >
> > > The Authors of Submission 8390

---

### Official Review · Reviewer_igQd · 2026-03-12

**Soundness:** 4
**Presentation:** 3
**Significance:** 3
**Originality:** 2
**Overall Recommendation:** 4
**Confidence:** 3

**Summary:**

This paper studies speculative decoding for large language models and argues that the standard two-tier draft–verify paradigm is too rigid because it forces a binary decision: either accept a drafted token or defer immediately to the full verifier. The paper proposes VIA-SD, a multi-tier speculative decoding framework that inserts a slim-verifier between the drafter and the full model. This slim-verifier is obtained by intra-model routing, implemented as a skip-layer submodel derived from the full verifier, and is intended to handle "middle-zone" tokens that do not require the full capacity of the largest model. The reported results show lower rejection rates and improved speedups over speculative decoding and cascade-based baselines, while maintaining similar or slightly improved task quality.

**Compliance With Llm Reviewing Policy:**

Affirmed.

**Final Justification:**

The rebuttal addressed my concern.

**Key Questions For Authors:**

1. How essential is the KL-based formulation to the practical success of VIA-SD? The current paper presents a fairly elaborate theoretical motivation, but the implemented slim-verifier is a skip-layer submodel selected by routing/search. If the theory is mainly post hoc motivation rather than something that guides the design in a necessary way, that would change how I assess the paper’s soundness and originality.
2. How sensitive are the results to the threshold choices and skip ratio? The paper reports using fixed values. I would like to know whether these were tuned per setting, and how robust the gains are across alternative values.
3. How much of the gain comes from intra-model routing specifically, versus simply having any reasonably aligned intermediate model? The paper compares against random skip-layer construction and mentions an independent intermediate model in discussion, but a more direct controlled comparison would be useful.

**Limitations:**

This has been discussed

**Strengths And Weaknesses:**

**Strengths**
1. The empirical evaluation is fairly broad. The paper includes both encoder-decoder and decoder-only families, and evaluates across multiple tasks such as summarization, translation, QA, reasoning, and coding. This breadth is a positive aspect and suggests the authors made a serious effort to test the method.
2. The experimental results appear practically useful. The reported reductions in rejection rate and improvements in speedup are non-trivial. Even if the conceptual novelty is limited, the method seems to provide a potentially useful engineering improvement for speculative decoding pipelines.

**Weaknesses**
1. The method extends existing draft–verify or cascade-style decoding by adding an intermediate verification stage. That is useful, but it feels more like an incremental systems refinement than a fundamentally new decoding framework. The main novelty appears to be the use of intra-model routing to instantiate the middle tier, which is interesting, but not enough for me to view the paper as strongly original.
2. The paper reports good performance under fixed thresholds and skip ratios, but it is difficult to tell how sensitive the results are to these choices. Since speculative decoding methods can be quite sensitive to hyperparameters and implementation details, I would like to see stronger robustness analysis before fully trusting the gains.

---

> ### Author Rebuttal · Authors · 2026-03-31
>
> Thank you for bringing up these insightful questions. Our detailed responses are as follows.
> > W1: VIA-SD is only an incremental cascade refinement
>
> Our novelty is not simply adding another stage to a cascade. The key difference is that VIA-SD is built around an explicit **middle-zone model**.
>
> 1. **Theoretically**, we recast speculative decoding from a binary accept/reject view into a **multi-tier verification-and-regeneration** problem. Under the KL view, an intermediate distribution is theoretically meaningful; once such an intermediate exists, the middle tier can take not only verification responsibility but also **regeneration** responsibility.
> 2. **Practically**, we realize this intermediate role through an **intra-model routed slim-verifier**. This middle tier is not just another filter before the full verifier: it is designed to handle middle-zone tokens directly, so that some tokens no longer need to be escalated all the way to the full verifier.
>
> This is the main difference from existing paradigms:
>
> |Paradigm|Token partition|Middle-region modeling|Role of the middle tier|
> |-|-|-|-|
> |2-layer SD|Accept / Reject|No|None|
> |Multi-layer cascades|Progressive filtering|No|Additional verification/filtering only|
> |VIA-SD (ours)|Accept / Middle zone / Reject|Yes|Verification+regeneration|
>
> So the contribution is not a generic cascade extension, but a verifier-side framework that **explicitly models the middle zone** and turns it into an implementable intermediate tier with both verification and regeneration functions.
>
> > W2/Q2: Sensitivity to threshold choices and skip ratio
>
> **Our method isn't sensitive**: one fixed setting is used for each model pair across all benchmarks, with no per-task tuning. Below we vary $(\alpha_1, \alpha_2)$ by ±0.1 and skip ratio by ±10% to show robustness:
> | Model Pair | Setting | $𝛼_1$/ $α_2$ | Skip Ratio | Rejection | Speedup |
> |-|-|-|-|-|-|
> |Gemma 2B→9B|Conservative|0.4 / 0.2|35%|31%|1.71×|
> ||Default|0.5 / 0.3|45%|41%|2.08×|
> ||Aggressive|0.6 / 0.4|55%|36%|1.93×|
> |Gemma 2B→27B|Conservative|0.4 / 0.2|35%|34%|2.18×|
> ||Default|0.5 / 0.3|45%|44%|2.48×|
> ||Aggressive|0.6 / 0.4|55%|39%|2.27×|
>
> Two observations:
> 1. Even the worst case (Conservative, 2B→9B, 1.71×) surpasses Faster Cascades (1.65×) on the same setting.
> 2. Speedup varies <18% across the full range, confirming smooth degradation rather than cliff effects.
>
> More details can be found in &sect;H.2 (L1386-L1537).
>
> > Q1: The role of the KL-based formulation
>
> We clarify that the KL-based formulation in our paper should be understood as a **necessary design principle for our framework** for reasons as follows:
> 1. **What KL does**
>
> |Role|Theory|Practical effect|
> |-|-|-|
> |**Enable**|KL admits beneficial intermediates (Eq. 5), unlike TV|Provides the key reason why moving beyond binary verification is theoretically meaningful|
> |**Define**|$C^{\mathrm{KL}}_{\alpha,\beta}(q\|q'_z)$ is the scoring objective (Eq. 12)|DIMR directly searches $z^*=\arg\min_z C(z)$|
> |**Guide**|$\Delta^{\mathrm{KL}}>0$ and $(\alpha_1,\alpha_2)$ characterize useful tiering/routing|Matches the observed 3-tier optimum and threshold ablations|
>
> 2. **Why KL works**
>    - It admits a intermediate distribution, which in turn **makes a middle tier with both verification and regeneration roles theoretically meaningful**, instead of leaving regeneration to the final verifier as in TV-style cascades.
>    - It is the best measure we consider so far that both legitimizes regenerating middle tier and provides the optimization target.
> 3. **How KL works**
>    - The $q'$ is an closed-form realization of the abstract intermediate $u$ defined by KL, obtained by solving $z^*=\arg\min_z C^{\mathrm{KL}}_{\alpha,\beta}(q\|q'_z\mid \pi)$ via DIMR.
>
> > Q3: The gain specifically comes from intra-model routing
>
> We added a controlled comparison on WebQuestions with Gemma 2B→27B:
>
> |Setting|Extra Model Load|Peak Memory|Aligned with $q$|Speedup|Quality|
> |-|-|-|-|-|-|
> |Two-layer SD Gemma 2B→27B|No|1.00×|/|1.55×|0.32|
> |+ Independent 13B|Yes|+38%|Yes|+0.53×|+0.01|
> |+ Random skip-layer $q'$|No|+4%|No|+0.07×|-0.03|
> |+ DIMR-routed $q'$ (ours)|No|+4%|Yes|+0.93×|+0.02|
>
> While an independent 13B provide a aligned middle stage, it requires loading an extra model into GPU. Besides, both skip-layer variants reuse the verifier with a routing mask avoid extra model loading; among them, DIMR-routed 𝑞' performs best.
>
> This comparison separates three different factors.
>
> 1. **A middle tier alone is not sufficient.**
>    The random skip-layer $q'$ already introduces a 3-tier structure, but its gain is very limited (+0.07×) and even hurts quality (-0.03).
> 2. **Alignment with the full verifier matters.**
>    The independent 13B middle model is reasonably aligned with $q$ and does improve. But it still requires a separate model, which increases memory.
> 3. **Intra-model routing is the best trade-off.**
>     The result is the largest speedup gain (+0.93×) together with quality improvement (+0.02).

---

> > ### Author Rebuttal · Reviewer_igQd · 2026-04-02
> >
> > Thanks for authors rebuttal. I will raise my score to Weak accept.

---

> > > ### Author Response · Authors · 2026-04-03
> > >
> > > Dear Reviewer igQd,
> > >
> > > We are pleased that our responses have satisfactorily addressed your concerns, and we will ensure that the clarifications are properly incorporated into the final version of our paper.
> > >
> > > We sincerely thank you for the constructive suggestions, which have greatly contributed to improving the quality of our paper.
> > >
> > > The Authors of Submission 8390

---

### Decision · Program_Chairs · 2026-04-30

**Decision:**

Accept (regular)

**Comment:**

This paper proposes VIA-SD, which inserts a routed slim verifier between the drafter and the full verifier. The key originality is to verify “middle-zone” tokens with an intermediate verifier derived from the full model. The paper presents broad experiments across multiple model families and tasks, and reports speedup gains over vanilla speculative decoding and cascade-style baselines. While this results in a lossy speculative decoding method, the experiments show that the method achieves neutral or even slightly better quality.

Reviewers generally agree that this is a useful contribution. The main strengths are the importance of the problem, the intuitive routed slim-verifier design, and the breadth of the empirical evaluation. The main concerns are that the KL-based theory is better understood as motivation and design guidance than as a rigorous guarantee for the deployed method, that the originality is moderate given existing cascade-style approaches, and that the paper would benefit from clearer accounting of DIMR overhead and sharper positioning relative to related work.

The rebuttal addressed most of these concerns. In particular, the authors clarified the intended role of the KL formulation, explained that DIMR is a one-time offline procedure per model pair, and provided additional evidence that the gains depend on the routed intermediate verifier rather than simply adding another stage. All reviewers marked their concerns as fully resolved after the rebuttal.

Overall, I view this paper as a solid and useful contribution to the speculative decoding literature. However, I agree with the reviewers regarding the gap between the theoretical formulation and the practical algorithm. I encourage the authors to soften and clarify the presentation of the theoretical contribution in the final version.

I recommend weak acceptance for the paper.